# A Negative Energy Balance Is Associated with Metabolic Dysfunctions in the Hypothalamus of a Humanized Preclinical Model of Alzheimer’s Disease, the 5XFAD Mouse

**DOI:** 10.3390/ijms22105365

**Published:** 2021-05-20

**Authors:** Antonio J. López-Gambero, Cristina Rosell-Valle, Dina Medina-Vera, Juan Antonio Navarro, Antonio Vargas, Patricia Rivera, Carlos Sanjuan, Fernando Rodríguez de Fonseca, Juan Suárez

**Affiliations:** 1Instituto de investigación Biomédica de Málaga-IBIMA, 29010 Málaga, Spain; antonio.lopez@ibima.eu (A.J.L.-G.); cristina.rosell@ibima.eu (C.R.-V.); dina.medina@ibima.eu (D.M.-V.); juan_naga@hotmail.es (J.A.N.); antonio.vargas@ibima.eu (A.V.); patricia.rivera@ibima.eu (P.R.); 2UGC Salud Mental, Hospital Regional Universitario de Málaga, 29010 Málaga, Spain; 3Departamento de Biología Celular, Genética y Fisiología, Campus de Teatinos s/n, Universidad de Málaga, Andalucia Tech, 29071 Málaga, Spain; 4Facultad de Medicina, Campus de Teatinos s/n, Universidad de Málaga, Andalucia Tech, 29071 Málaga, Spain; 5UGC Corazón, Hospital Universitario Virgen de la Victoria, 29010 Málaga, Spain; 6EURONUTRA S.L, Parque Tecnológico de Andalucía, 29590 Campanillas, Spain; euronutra@euronutra.eu; 7Departamento de Anatomía Humana, Medicina Legal e Historia de la Ciencia, Universidad de Málaga, 29071 Málaga, Spain

**Keywords:** Alzheimer’s disease, 5xFAD, insulin signaling, energy expenditure, hypothalamus, neuroinflammation

## Abstract

Increasing evidence links metabolic disorders with neurodegenerative processes including Alzheimer’s disease (AD). Late AD is associated with amyloid (Aβ) plaque accumulation, neuroinflammation, and central insulin resistance. Here, a humanized AD model, the 5xFAD mouse model, was used to further explore food intake, energy expenditure, neuroinflammation, and neuroendocrine signaling in the hypothalamus. Experiments were performed on 6-month-old male and female full transgenic (Tg^5xFAD/5xFAD^), heterozygous (Tg^5xFAD/-^), and non-transgenic (Non-Tg) littermates. Although histological analysis showed absence of Aβ plaques in the hypothalamus of 5xFAD mice, this brain region displayed increased protein levels of GFAP and IBA1 in both Tg^5xFAD/-^ and Tg^5xFAD/5xFAD^ mice and increased expression of IL-1β in Tg^5xFAD/5xFAD^ mice, suggesting neuroinflammation. This condition was accompanied by decreased body weight, food intake, and energy expenditure in both Tg^5xFAD/-^ and Tg^5xFAD/5xFAD^ mice. Negative energy balance was associated with altered circulating levels of insulin, GLP-1, GIP, ghrelin, and resistin; decreased insulin and leptin hypothalamic signaling; dysregulation in main metabolic sensors (phosphorylated IRS1, STAT5, AMPK, mTOR, ERK2); and neuropeptides controlling energy balance (NPY, AgRP, orexin, MCH). These results suggest that glial activation and metabolic dysfunctions in the hypothalamus of a mouse model of AD likely result in negative energy balance, which may contribute to AD pathogenesis development.

## 1. Introduction

Physical decline and impairment of metabolic homeostasis are the main features of the human aging process [1]. The dysregulation of the metabolic network leads to an age-related elevated risk of suffering from chronic metabolic disorders, especially insulin resistance-related pathologies. In addition to the well-known peripheral role for insulin on glucose and energy storage, insulin also regulates a series of cognitive processes such as memory formation through its effects on glial–neuronal metabolic coupling. Central insulin resistance is a common feature linked to premature aging and observed in neurological disorders, including early stages of Alzheimer’s disease (AD) [2].

AD dementia is the main form of dementia in humans. AD is a progressive neurological disorder with a mostly sporadic origin that is characterized by the loss of cognitive functions such as memory, reasoning, or language, leading to death at about 3–9 years after diagnosis. Common features of AD are aberrant production and deposition of β-amyloid (Aβ) peptides, either in Aβ40 or Aβ42 fragments and Tau protein hyperphosphorylation aggregates (neurofibrillary tangles), disruption of normal autophagic processes, generation of reactive oxygen species (ROS) and reactive nitrogen species (RNS), and cell death [3,4]. The combination of the above factors is seen be cause for cognitive impairment and memory loss. Recent studies suggest that unhealthy dietary habits and microbiota changes mediate neuroinflammation, modify neurovascular coupling, and create metabolic disturbances deriving on oxidate/nitrosative stress and insulin resistance. All these factors have been identified to contribute to the cognitive decline in AD patients.

Studies have shown AD progression is linked to dysregulated insulin signaling in the frontal cortex and hippocampus, and postmortem analysis of human hippocampal tissue shows correlation between high serine-inhibitory phosphorylation of insulin receptor substrate 1 (IRS1) and those of oligomeric Aβ plaques, which were negatively associated with working memory and episodic memory [5]. Further studies have shown that early hyperactivation of insulin signaling may cause negative feedback mediated by energy sensors such as mTOR, thus leading to impaired metabolic sensing in the neuronal population [6,7].

Aside from metabolic impairment, epidemiological studies have long appreciated weight loss associated with development of neurodegenerative diseases such as AD in the elderly [8]. Clinical data shows that weight loss precedes development of dementia in AD and may be a preclinical indicator of the disease [9]. Increased presence of AD markers (Aβ and total Tau) in cerebrospinal fluid has been correlated to decreased body mass index (BMI), which reinforces the link between body weight and metabolic disturbances with development of dementia and AD [10]. This negative energy status in AD patients with disturbed insulin signaling in the brain may exacerbate AD pathology. However, despite the age-associated frailty, the molecular and physiological mechanisms deriving from weight loss and metabolic impairment in AD patients are still poorly described, and data on neuroendocrine changes preceding AD are scarce.

The hypothalamus is the key regulator of metabolic processes in the organism. Food intake and energy expenditure are controlled by neuronal populations secreting orexigenic neuropeptides NPY and AgRP (responsible for promoting appetite and decreasing metabolic rate) and anorexigenic neuropeptides POMC and CART (deriving on satiety and promoting energy expenditure). Studies in AD mouse model 3xtg, which enhances Tau hyperphosphorylation, have shown inflammation and loss of neuronal population controlling energy expenditure and food intake such as POMC- and NPY-expressing neurons in 6-month-old mice [11]. Other studies have shown a hypermetabolic state in 3xtg mice, as observed by increased food intake and energy expenditure in young mice, whereas energy expenditure is significantly decreased in 3xtg mice at 18 months old, accompanied by decreased body weight [12]. NPY electrophysiological responses were also observed in another mouse model of AD overexpressing the Swedish mutation of APP (Tg2576) [13]. Studies in mice have helped gain insight into how hypothalamic dysfunction and metabolic disturbances play an early role in AD development. However, exact changes and molecular mechanisms underlying these events are still to be discerned.

Changes in brain metabolism in humans have been observed to be distinct from males and females, as the female brain undergoes earlier signals of hypometabolic status preceding and stating susceptibility in developing cognitive decline and AD [14]. Given differences in brain metabolic rate between males and females, sex differences should also be regarded as a main factor in the development of AD. However, there is a lack of studies on sex differences regarding the use of humanized preclinical models of AD.

A relevant issue on the clinical approach to AD and related pathologies that lead to cognitive impairment is the fact that most of the research effort on therapeutics have focused on either fighting the symptoms by boosting certain deteriorated transmission pathways (as it is the example of anti-acetylcholinesterase drugs to enhance cholinergic transmission) or reducing Aβ load via immunotherapy. However, there is a clear lack of therapeutic development designed to restore metabolic impairments associated with these neurodegenerative disorders. In our study, we aimed to investigate food intake, energy expenditure, and neuroinflammation and the mechanisms regulating these processes in the hypothalami of 6-month-old male and female 5xFAD (FAD: family Alzheimer’s disease) transgenic mouse model of AD, which is one of the most early onset AD models of neurodegeneration and amyloid pathology, where cognitive decline starts by age of 4 months increases through age [15,16,17]. Biomarkers of neuroinflammation, metabolic-regulating hormones, insulin signaling, energy sensors, and neuropeptide levels were assessed in an attempt to gain insight into how energy balance is disrupted, contributing to development of AD pathology.

## 2. Results

### 2.1. Heterozygous (Tg^5xFAD/-^) and Homozygous (Tg^5xFAD/5xFAD^) Transgenic 5xFAD Mice Showed Decreased Body Weight, Food Intake, and Energy Expenditure at 6 Months

Since the first signs of glucose metabolism decline in 5xFAD mice brain regions are observed at 6 months [18], we assessed body weight and metabolic parameters in 5xFAD at this age. At 6 months old, two-way ANOVA showed no interactions between genotype and sex (F_2,43_ = 2.766; *p* = 0.0741), but a main effect of sex (F_1,43_ = 120.6; *p* < 0.001) and genotype F_2,43_ = 5.182; *p* = 0.0096) (Figure 1A). Multiple comparisons by Tukey’s test showed decreased body weight in male Tg^5xFAD/5xFAD^ as compared to non-Tg (*p* < 0.01) and Tg^5xFAD/-^ (*p* < 0.01). Both female Tg^5xFAD/5xFAD^ and Tg^5xFAD/-^ also showed significant decrease in body weight as compared to non-Tg females (*p* < 0.05) (Figure 1A).

Next, we aimed to determine if these defects in body weight were related to changes in food intake patterns and energy expenditure (EE) for 48 h as they were placed in metabolic cages. Two-way ANOVA showed the main genotype effect on food intake per body weight during light phase (day) (F_2,47_ = 4.096; *p* = 0.0229) (Figure 1B) and dark phase (night) (F_2,48_ = 15.44; *p* < 0.001) (Figure 1C). Male Tg^5xFAD/5xFAD^ showed a slight increase in food intake as compared to Tg^5xFAD/-^ mice during the day (Tukey’s test: *p* < 0.05) (Figure 1B). However, during the night, male Tg^5xFAD/5xFAD^ showed a great decrease in food intake as compared to non-Tg (Tukey’s test: *p* < 0.001) (Figure 1C). Both female Tg^5xFAD/-^ and Tg^5xFAD/5xFAD^ mice also showed decreased food intake as compared to non-Tg females (Tukey’s test: *p* < 0.01 and *p* < 0.001, respectively) (Figure 1C).

EE is calculated by indirect calorimetry, using VO_2_ and VCO_2_ flow rates and normalized to body weight (BW) and raised to the power 0.75, since small animals such as mice show greater basal energy expenditure than larger animals such as humans (Figure 1D–G). No interaction effect was observed by two-way ANOVA, but there was a main effect of genotype during the day (F_2,44_ = 3.227; *p* = 0.0492) and night (F_2,44_ = 3.661; *p* = 0.0338), as well as a sex main effect during the day (F_1,44_ = 17.41; *p* = 0.001) and night (F_1,44_ = 17.42; *p* = 0.001), as female mice usually have a greater metabolic rate. Tukey’s test showed only Tg^5xFAD/5xFAD^ mice had slight decreases in EE, as shown by male mice during the day (*p* < 0.05 vs. non-Tg group), as well as being more pronounced in female mice during the day (*p* < 0.05 vs. Non-Tg and Tg^5xFAD/-^ groups) and night (*p* < 0.05 vs. Non-Tg and Tg^5xFAD/-^ groups). Pattern of activity and rearing was also different between males and females during the night (Appendix A), as shown by main sex effect (F_1,64_ = 8.305, *p* = 0.0054 for night activity; F_1,64_ = 16.41, *p* = 0.001 for night rearing), showing decreased night activity in Tg^5xFAD/5xFAD^ females compared to non-Tg females (*p* < 0.05) (Appendix A).

Respiratory quotient (RQ) as calculated by VCO_2_/VO_2_ indicates rate of glucose or fat fuel utilization, since lower RQ measurements correlate to higher O_2_ consumption in lipid β-oxidation (Figure 1H–K). Two-way ANOVA analysis determined only sex effect during the day (F_1,44_ = 7.176; *p* = 0.0104). However, both female Tg^5xFAD/-^ and Tg^5xFAD/5xFAD^ mice presented lower RQ during the night (Tukey’s test; *p* < 0.05 vs. non-Tg group) (Figure 1K).

### 2.2. Heterozygous (Tg^5xFAD/-^) and Homozygous (Tg^5xFAD/5xFAD^) Transgenic 5xFAD Mice Showed Increased Hypothalamic Tau Phosphorylation and Inflammation Despite Absence of Aβ Plaques in the Hypothalamus

We investigated if changes in body weight and metabolic parameters were related to abnormalities in the hypothalamus, as it is the main brain region responsible for control of energy homeostasis [19]. There were no visible Aβ plaques in the hypothalamus of none of the three genotypes (Figure 2A–F). However, the presence of Aβ plaques was significantly found in the hippocampus of both Tg^5xFAD/-^ and Tg^5xFAD/5xFAD^ male and female mice (Figure 2G), indicating amyloid pathology affecting the brain region responsible for learning and memory, exhibiting a projection to the hypothalamus via fimbria-fornix bundle.

Despite the absence of Aβ plaques, as observed by immunohistochemical sections, we aimed to determine by Western blotting if there were changes in Tau protein activity in the hypothalamus, as its hyperphoshorylation that leads to formation of toxic microtubule neurofibrillary tangles (NFTs), a hallmark of AD [5]. We also evaluated changes in protein levels of Tau kinases (Figure 2H–N). Tau is seen to be a target of the protein kinase CDK5, a member of the cyclin-dependent kinases (Cdks). CDK5 is activated physiologically via p35. However, in pathological conditions, as observed in AD, CDK5 is hyperactivated by p25, a truncated form of p35, leading to hyperphosphorylation [20].

Two-way ANOVA analysis revealed a significant main effect of genotype in Tau-AT8 phosphorylation (F_2,28_ = 5.22; *p* = 0.0118), CDK5 protein levels (F_2,28_ = 6.353; *p* = 0.0053), and p25 protein levels (F_2,28_ = 10.72; *p* < 0.001), but also main sex effect in CDK5 (F_1,28_ = 32.17; *p* < 0.001) and p25 protein levels (F_1,28_ = 17.04; *p* < 0.001). Interaction was also observed in CDK5 (F_2,28_ = 6.438; *p* = 0.005). Multiple comparison showed increased Tau-AT8 hyperphosphorylation in both male and female Tg^5xFAD/5xFAD^ compared to Non-Tg (*p* < 0.05) (Figure 2G). Main sex effect was observed in kinase activity in females, as both Tg^5xFAD/-^ and Tg^5xFAD/5xFAD^ female mice showed increased CDK5 protein levels (Tukey’s test; *p* < 0.001 and *p* < 0.05, respectively) and p25 (*p* < 0.001 and *p* < 0.05, respectively) as compared to Non-Tg females (Figure 2I,L). These results suggest male hyperactivation of Tau may be regulated by different kinase activities. Intriguingly, despite high CDK5 and p25 protein levels in heterozygous Tg^5xFAD/-^ females, no significant increase in Tau activity or protein levels were observed (Figure 2H).

We also evaluated protein expression of LRP1, a receptor that mediates Aβ internalization and degradation in the brain, as well as Aβ clearance across the blood–brain barrier (BBB) [21]. Intriguingly, LRP1 levels were significantly increased in Tg^5xFAD/-^ (Figure 2M) but not Tg^5xFAD/5xFAD^ male mice (Tukey’s test; *p* < 0.01), and non-significantly in Tg^5xFAD/-^ female mice (*p* = 0.146). These results suggest heterozygous 5xFAD mice could exhibit a more marked Aβ clearance as compared to homozygous 5xFAD mice.

As a means to assess pathogenicity in the hypothalamus, we determined mRNA and protein expression of inflammatory cytokine neuroinflammatory markers (Figure 3A–G). Two-way ANOVA analysis showed the main genotype effect on *Tnf* (F_2,31_ = 6.73; *p* = 0.0037) and *Il1b* (F_2,31_ = 7.198; *p* = 0.0027) mRNA levels, but no significant changes in expression of *Il6*. It should be noted that inflammatory response observed was different between genotypes, since the expression of *Tnf* was significantly elevated in Tg^5xFAD/-^ mice but not in Tg^5xFAD/5xFAD^ and, on the contrary, that of *Il1b* was augmented in Tg^5xFAD/5xFAD^ but not in Tg^5xFAD/-^ as compared to Non-Tg mice (Figure 3A,B). Protein expression of glia (GFAP) and microglia/macrophage (IBA1) markers was also significantly elevated in Tg^5xFAD/-^ and Tg^5xFAD/5xFAD^ mice (Figure 3D,F). However, Il-1β protein levels were only significantly increased in female Tg^5xFAD/5xFAD^ mice (Figure 3E). These results showcase that both heterozygous Tg^5xFAD/-^ and homozygous Tg^5xFAD/5xFAD^ showed an inflammatory response in the hypothalamus, despite absence of Aβ immunoreactivity, with little differences between males and females and a different pattern of cytokine release profile.

Correlation test was run to assess if neuroinflammation was directly associated with decreased body weight in 5xFAD mice (Table 1). We found no correlation between body weight and neuroinflammatory marker levels (IL-1β, TNF-α, GFAP, and IBA1), which were significantly elevated in male mice of either genotype. In Non-Tg females, body weight was negatively correlated to TNF-α (*p* < 0.05) and IBA1 (*p* < 0.05). In Tg^5xFAD/-^ females, IL-1β (*p* < 0.05) and IBA1 (*p* < 0.05) were also related to body weight. Tg^5xFAD/5xFAD^ female body weights showed more associations with IL-1β (*p* < 0.05), TNF-α (*p* < 0.05), and IBA1 (*p* < 0.05).

### 2.3. Transgenic 5xFAD Mice Had Low Circulating Levels of Insulin, GLP-1, Ghrelin, and Resistin, as Well as Altered Activity and Expression of Hormone Receptors in the Hypothalamus

Brain insulin resistance is a pattern observed in mice models of AD. Previous studies have observed low plasma levels of leptin in a different mouse model of AD [13]. In addition, the pancreas expresses amyloid precursor protein (APP), and there is a suspicion of a contribution of Aβ-induced damage in the pancreas to insulin resistance-linked alterations in AD [22]. Thus, we wanted to determine if metabolic dysfunction was associated with either increased or decreased plasma levels of insulin, insulin secretagogues, or adipokines, as well as altered hormone receptor activity in the hypothalamus, in order to explain metabolic changes in these animals. We found significant genotype differences in basal plasma levels of insulin (two-way ANOVA; F_2,33_ = 10.26; *p* < 0.001), with a marked decrease in plasma insulin Tg^5xFAD/-^ and Tg^5xFAD/5xFAD^ mice (Figure 4A). These results were accompanied by decreased mRNA expression of insulin receptor (*Insr*) in female Tg^5xFAD/-^ and Tg^5xFAD/5xFAD^ mice (Figure 4D). Activity of insulin receptor as measured by tyrosine phosphorylation (p-IR/IR protein levels ratio) was not significantly different between genotypes, although a non-significant decrease was observed again in female Tg^5xFAD/-^ and Tg^5xFAD/5xFAD^ mice (Figure 4G,J).

GLP-1 is known to promote glucose-stimulated insulin release, but its effects in absence of glucose bolus are under discussion [23]. Nevertheless, basal insulin levels correlate to those of basal GLP-1, and decreased GLP-1 plasma is observed after weight loss [24]. Interestingly, plasma levels of GLP-1 were decreased in female Tg^5xFAD/5xFAD^ mice (Figure 4B). Regarding GLP-1 activity, according to insulin and insulin receptor levels, we also observed decreased GLP-1 receptor (*Glp1r*) mRNA expression in the hypothalamus of Tg^5xFAD/-^ and Tg^5xFAD/5xFAD^ mice (Figure 4E) with a significant genotype main effect (F_1,33_ = 8.042; *p* = 0.0078) and a significant decrease in GLP1-R protein levels in male Tg^5xFAD/5xFAD^ mice as compared to Tg^5xFAD/-^ and non-Tg male mice (Figure 4H,J). These results over GLP-1 activity in the hypothalamus may agree with an impaired insulin response. Despite this, we also observed decreased plasma levels of ghrelin in female Tg^5xFAD/5xFAD^ mice (Figure 4C). Ghrelin reduces glucose-stimulated insulin secretion, and thus ghrelin levels would be expected to be either unchanged or slightly increased [25]. Since ghrelin is secreted in an autocrine fashion in the hypothalamus, we measured ghrelin (*Ghrl*) mRNA levels and found a slight increase in male Tg^5xFAD/5xFAD^ mice as compared to Tg^5xFAD/-^ mice (Figure 4F). However, ghrelin receptor (*Ghsr*) mRNA levels were significantly decreased in male but not female Tg^5xFAD/-^ and Tg^5xFAD/5xFAD^ mice (Figure 4I).

Since Tg^5xFAD/-^ and Tg^5xFAD/5xFAD^ mice show a leaner phenotype, we also measured profile of adipokines leptin and resistin. Basal levels of both hormones are normally elevated in obese mice. Sexual dimorphism is also seen, with plasma leptin being increased in male mice, whereas plasma resistin is higher in female mice [26]. In accordance with this, we found significant sex differences in plasma leptin (two-way ANOVA; F_1,33_ = 8.826; *p* = 0.0055) and resistin (F_1,33_ = 29.36; *p* < 0.001) and also genotype differences in resistin (F_2,33_ = 9.612; *p* < 0.001) (Figure 5A,B).

Despite a leaner phenotype, no changes in leptin receptor mRNA expression or protein activity (as determined by Tyr1077 phosphorylation) were observed, although there was a non-significant trend towards lower receptor signaling in the hypothalamus of Tg^5xFAD/-^ and Tg^5xFAD/5xFAD^ female mice (Figure 5D,G,H). Resistin is regarded as a promoter of obesity-mediated hypothalamic inflammation via TLR4 receptor, being correlated to insulin resistance [27]. It was surprising to find decreased plasma levels of resistin in both male and female Tg^5xFAD/-^ mice as compared to non-Tg mice (Figure 5B). We also measured central resistin (*Retn*) mRNA expression, which showed a marked decrease in female Tg^5xFAD/-^ and Tg^5xFAD/5xFAD^ mice as compared to non-Tg (*p* < 0.05 and *p* < 0.05, respectively) (Figure 5E). These results were not accompanied by changes on TLR4 (*Tlr4*) mRNA, but increased TLR4 protein levels were also observed in female Tg^5xFAD/-^ (*p* < 0.001) and Tg^5xFAD/5xFAD^ (*p* < 0.05) mice (Figure 5C,F,H). It is seen that autocrine or central resistin infusion interferes in food intake and peripheral insulin sensitivity fatty acid synthesis via modulation of neuropeptide NPY expression [28]. Changes in hypothalamic resistin expression and TLR4 levels may be more in accordance with a metabolic rather than inflammatory role in whole-body energy homeostasis.

Along with GLP-1, GIP also promotes glucose-stimulated insulin release. We found a main genotype effect for decreased GIP plasma levels (F_2,33_ = 3.394; *p* = 0.0457). GIP levels were lower in male Tg^5xFAD/-^ and Tg^5xFAD/5xFAD^ mice compared to non-Tg, but changes did not reach statistical significance in Tg^5xFAD/-^ mice (*p* = 0.010 and *p* < 0.05, respectively) (Appendix A). Glucagon and PAI-1 levels were, however, not significantly different between groups (Appendix A).

Correlation test was run to assess if abnormal plasma and hypothalamic levels of hormones were related to decreased body weight in 5xFAD mice (Table 2). We found insulin levels were also directly related to body weight in Tg^5xFAD/-^ males (*p* < 0.05), Tg^5xFAD/5xFAD^ males (*p* < 0.01), Tg^5xFAD/-^ females (*p* < 0.05), and Tg^5xFAD/5xFAD^ females (*p* < 0.05). Leptin was also positively related to body weight only in Tg^5xFAD/-^ males (*p* < 0.001) and Tg^5xFAD/5xFAD^ males (*p* < 0.05), but not females, indicating low leptin levels were effectively lower with decreased body weight. Plasma levels of resistin were positively related to body weight in Tg^5xFAD/5xFAD^ males (*p* < 0.05), Tg^5xFAD/-^ females (*p* < 0.01), and Tg^5xFAD/5xFAD^ females (*p* < 0.05). Hypothalamic resistin expression was also related to decreased body weight in Tg^5xFAD/-^ females (*p* < 0.01) and Tg^5xFAD/5xFAD^ females (*p* < 0.05). As for hormones regulating insulin secretion, ghrelin and GLP-1 were positively associated with body weight in Tg^5xFAD/5xFAD^ females (*p* < 0.05) (*p* < 0.05 for both hormones), whereas GIP was positively associated with body weight in Tg^5xFAD/-^ males (*p* < 0.05) and Tg^5xFAD/5xFAD^ males (*p* < 0.05). Plasma levels of glucagon were not associated with body weight in any cases.

Correlation tests were also run to assess if decreased insulin levels were related to decreased incretins (GLP-1 and GIP) plasma levels or increased ghrelin and glucagon levels (Table 3). Interestingly, decreased plasma levels of insulin were differently associated in males and females, with insulin being positively associated with GIP in Tg^5xFAD/-^ males (*p* < 0.05) and Tg^5xFAD/5xFAD^ males (*p* < 0.05), whereas it was associated with plasma levels of GLP-1 in Tg^5xFAD/5xFAD^ females (*p* < 0.01). No correlation was observed for plasma levels of glucagon or ghrelin.

### 2.4. Insulin and Leptin Signaling Was Decreased in Hypothalamus of Transgenic 5xFAD Mice at 6 Months of Age

The classical mechanisms of action in insulin and leptin signaling have been extensively described and reviewed [29,30]. The binding of insulin to its receptor (IR) in target tissues promotes tyrosine autophosphorylation, recruiting IR substrates as IRS1, whereas hypothalamic leptin receptor (LepR) actions modulating metabolism include the JAK-STAT3 and STAT5 signaling.

Since IRS1 activity is controlled by inhibitory and activating phosphorylations on serine and tyrosine residues, respectively, we determined the ratio of activity by determining the pTyr892/pSer612 ratio and total protein levels (Figure 6A,D,G). There was a main genotype effect to decrease IRS phosphorylation activity in 5xFAD mice (two-way ANOVA; F_2,28_ = 8.865; *p* = 0.001). Tukey’s test also revealed significant decreased pTyr892/pSer612 ratio in Tg^5xFAD/-^ and Tg^5xFAD/5xFAD^ male mice compared to non-Tg males (*p* < 0.05; *p* < 0.01, respectively) and also in Tg^5xFAD/5xFAD^ female mice compared to both Tg^5xFAD/-^ and non-Tg females (*p* < 0.05; *p* < 0.05, respectively) (Figure 6A). There was also a main genotype effect observed in decreased IRS1 protein levels (F_2,28_ = 6.404; *p* = 0.0051) (Figure 6D).

We also determined activity and total protein level LepR signaling elements STAT5 and STAT3 (Figure 6B,C,E–G). Phosphorylation of Tyr1077 on LepR, which we measured, regulates primarily STAT5 [31]. In a similar trend to what we observed with pTyr1077 on LepR, there was a main genotype effect in a decreased STAT5 activating phosphorylation (F_2,28_ = 4.752; *p* = 0.0167). Again, Tukey’s test revealed particular significant STAT5 phosphorylation in both Tg^5xFAD/-^ and Tg^5xFAD/5xFAD^ female mice compared to non-Tg females (*p* < 0.05; *p* < 0.01, respectively) (Figure 6B). Intriguingly, there were no changes in STAT3 phosphorylation (Figure 6C,G), but we observed a main sex and genotype effect and interaction in increased STAT3 protein levels (two-way ANOVA; F_1,28_ = 20.78, *p* < 0.001; F_2,28_ = 14.32, *p* < 0.001; F_2,28_ = 14.78, *p* < 0.001, respectively). Tukey’s test showed Tg^5xFAD/5xFAD^ males had higher STAT3 protein levels than Tg^5xFAD/-^ (*p* < 0.001) and non-Tg (*p* < 0.01) males, whereas Tg^5xFAD/-^ and Tg^5xFAD/5xFAD^ females had lower STAT3 protein levels compared to non-Tg females (*p* < 0.001; *p* < 0.001, respectively) (Figure 6F,G).

### 2.5. Hypothalamic Energy Sensors Were Dysregulated in Transgenic 5xFAD Mice at 6 Months of Age

Energy sensors in the hypothalamus, such as AMPK, mTOR, and ERK proteins, interplay in hormone receptor signaling pathways and regulate processes of food intake, glucose uptake, mitochondrial metabolism, and whole-body energy homeostasis [32,33,34]. Normally, mTOR acts as a pro-anabolic factor in response to insulin signaling. AMPK, in contrast, counteracts mTOR activity in fasting state. Changes in these energy sensors in the hypothalamus could help interpretation of negative energy balance in transgenic 5xFAD mice (Figure 7).

Two-way ANOVA analysis showed a main effect of genotype for increased phosphorylation of AMPK in 5xFAD mice (F_2,28_ = 5.317; *p* = 0.011), but also a main sex effect for increased pAMPK in females (F_1,28_ = 14.9; *p* < 0.001) and interaction “sex x genotype” (F_2,28_ = 4.275; *p* = 0.024). This is resumed in a particular significant increase in pAMPK only on Tg^5xFAD/5xFAD^ males compared to Tg^5xFAD/-^ (*p* < 0.001) and non-Tg males (*p* < 0.001) (Figure 7A,I). There were also significant main effects of sex (F_1,14_ = 5.665; *p* = 0.0321), genotype (F_2,14_ = 4.392; *p* = 0.0331), and interaction “sex x genotype” in AMPK total protein levels (F_2,14_ = 6.305; *p* = 0.0112). Tukey’s test showed Tg^5xFAD/5xFAD^ female mice had increased AMPK protein levels compared to Tg (*p* < 0.05) and non-Tg females (*p* < 0.001) (Figure 7D,I). As expected, phosphorylation of mTOR showed a contrary trend, with a main genotype effect for decreased pmTOR (F_2,28_ = 3.399; *p* = 0.0477). In particular, female Tg^5xFAD/5xFAD^ had decreased pmTOR compared to non-Tg females (*p* < 0.05) (Figure 7B,I). mTOR total protein levels were generally increased in 5xFAD mice, as observed with main genotype effect (two-way ANOVA; F_2,28_ = 10.49; *p* < 0.001). This was also shown in significant differences between mTOR total protein in Tg^5xFAD/5xFAD^ males compared to Tg^5xFAD/-^ (*p* < 0.05) and non-Tg males (*p* < 0.05), and also in Tg^5xFAD/5xFAD^ females compared to Tg^5xFAD/-^ (*p* < 0.05) and non-Tg females (*p* < 0.05) (Figure 7E,I).

As for ERK activity, there was a significant main effect of sex showing increased pERK1 (F_1,28_ = 14.95; *p* < 0.001) and pERK2 (F_1,28_ = 11.77; *p* = 0.0019) in males. Tukey’s test only showed a particular increase in pERK2 in Tg^5xFAD/-^ males compared to Tg^5xFAD/5xFAD^ males (*p* < 0.01) and non-Tg males (*p* < 0.05) (Figure 7G,I). There was an interaction of “sex x genotype” in ERK1 and ERK2 total protein levels (F_2,28_ = 4.661, *p* = 0.0179; F_2,28_ = 4.88, *p* = 0.0152, respectively), showing a trend towards decreased total ERK1 and ERK2 in 5xFAD males and increase in 5xFAD females. Tukey’s test also showed increased ERK1 protein levels in Tg^5xFAD/5xFAD^ females compared to non-Tg females (*p* < 0.05) and the same trend for ERK2 protein levels in Tg^5xFAD/5xFAD^ females compared to non-Tg females (*p* < 0.001) (Figure 7F,H).

### 2.6. A Decrease in Orexigenic Neuropeptides Was Observed in Male Transgenic 5xFAD Mice at 6 Months of Age

Control of energy homeostasis in the hypothalamus resides ultimately in the interplay of distinct neuropeptides NPY and AgRP, to promote body mass increase and suppress appetite; POMC and CART, which cause weight loss by inhibiting food intake and stimulating energy expenditure; and others such as orexin and MCH, linking appetite patterns to circadian rhythms, with orexin increasing appetite and wakefulness in a food-seeking context, and MCH stimulating consumption of palatable food and promoting paradoxical sleep [19,35,36,37]. Balance in the secretion of these neuropeptides is essential in controlling behavior intake and whole-energy homeostasis. We determined mRNA expression of these neuropeptides in the whole hypothalamus.

Two-way ANOVA showed that there was a main effect of genotype to decrease NPY hypothalamic mRNA expression (*Npy*) (F_2,31_ = 4.511; *p* = 0.0191) and also an interaction in “sex x genotype”, revealing decreased NPY primarily in males (F_2,31_ = 3.479; *p* = 0.0434). Tukey’s test showed this decrease was more pronounced in Tg^5xFAD/-^ males than Tg^5xFAD/5xFAD^ males as compared to non-Tg males (*p* < 0.01; *p* < 0.05, respectively) (Figure 8A). AgRP mRNA expression (*AgRP*) showed a similar trend as NPY, but no significant main effects nor particular differences were observed (Figure 8B). Contrary to orexigenic neuropeptides, no overall changes in mRNA expression of anorexigenic neuropeptides POMC (*Pomc*) and CART (*Cartpt*) were observed (Figure 8C,D). Intriguingly, as opposed to NPY, orexin mRNA expression (*Hcrt*) was significantly increased in Tg^5xFAD/-^ males compared to non-Tg males (*p* < 0.01) (Figure 8E). There was also a main genotype effect in an increase in MCH mRNA expression (*Pmch*) in 5xFAD mice (F_2,31_ = 3.317; *p* = 0.0495) with a significant increase in Tg^5xFAD/5xFAD^ females compared to non-Tg females (*p* < 0.05) (Figure 8F).

Correlation tests were run to assess if neuropeptide levels were directly associated with cytokines that were significantly increased in 5xFAD mice (IL-1β and TNF-α) (Table 4). POMC levels were negatively associated with IL-1β in Tg^5xFAD/5xFAD^ females (*p* < 0.05). MCH was associated with IL-1β in non-Tg males (*p* < 0.01), but not in transgenic 5xFAD mice. Interestingly, hypothalamic resistin was negatively associated with IL-1β and TNF-α differently in males and females. Resistin was negatively correlated to IL-1β in Tg^5xFAD/-^ females (*p* < 0.01) and Tg^5xFAD/5xFAD^ females (*p* < 0.05), and also to TNF-α in Tg^5xFAD/5xFAD^ females (*p* < 0.01). In male mice, resistin was not significantly associated with IL-1β, but correlation was observed with TNF-α in Tg^5xFAD/-^ males (*p* < 0.05) and Tg^5xFAD/5xFAD^ males (*p* < 0.01). Since NPY dysfunction has been observed in AD mice [13], we also ran a correlation test to determine if NPY expression was accompanied with expression of other neuropeptides. NPY levels were associated to AgRP in all groups but Non-Tg males, as they are expressed in the same neuronal population in the arcuate nucleus of the hypothalamus (ARC). Interestingly, NPY levels were also related to hypothalamic POMC in in Tg^5xFAD/-^ females (*p* < 0.05) and Tg^5xFAD/5xFAD^ females (*p* < 0.05). NPY levels were also related similarly to orexin (HCRT) and MCH in Tg^5xFAD/-^ males, Tg^5xFAD/5xFAD^ males, and Tg^5xFAD/5xFAD^ females (*p* < 0.05 for all).

### 2.7. Lipid Plasma Profile Was Altered in Female 5xFAD Mice

Plasma lipid profile in 5xFAD mice was determined in order to assess if AD metabolic dysfunction was related to altered lipid metabolism as previously observed in AD patients [38,39]. We determined levels of plasma PAI-1, as it is associated with increased risk of thrombolysis (Appendix A). There was a significant sex effect in PAI-1 levels (F_1,33_ = 8.537; *p* = 0.0062), with a tendency to increased PAI-1 plasma levels in female 5xFAD mice that did not reach statistical significance. Biochemistry showed altered total cholesterol, HDL, LDL, and triglyceride plasma levels and also hepatic glutamic oxaloacetic transaminase (GOT) (Appendix A). Two-way ANOVA showed interaction in “sex x genotype” in total cholesterol (F_2,42_ = 3.39; *p* = 0.0432). Breaking down cholesterol results, we observed for HDL a main effect of genotype (F_2,42_ = 5.653; *p* = 0.0067), sex (F_1,42_ = 16.64; *p* = 0.0002), and interaction sex x genotype (F_2,42_ = 3.577; *p* = 0.0368), while in LDL there was also a main effect in genotype (F_2,42_ = 3.331; *p* = 0.0454) and sex (F_1,42_ = 5.56; *p* = 0.0231). Triglyceride levels showed an interaction “sex x genotype” (F_2,42_ = 3.275; *p* = 0.0477). GOT was distinctly affected by genotype (F_2,42_ = 6.773; *p* = 0.0028). Total cholesterol was significantly decreased in Tg^5xFAD/-^ females compared to non-Tg females (*p* < 0.01). Conversely, HDL levels were significantly increased in Tg^5xFAD/5xFAD^ females compared to both non-Tg and Tg^5xFAD/-^ females (*p* < 0.01 for both) (Appendix A), whereas LDL was significantly decreased in Tg^5xFAD/5xFAD^ females also compared to both non-Tg and Tg^5xFAD/-^ females (*p* < 0.05 for both) (Appendix A). GOT levels were significantly increased in Tg^5xFAD/5xFAD^ females compared to Tg^5xFAD/-^ females (*p* < 0.05) and non-Tg females (*p* < 0.01) (Appendix A).

To assess if altered lipid plasma levels could be affected by white adipose tissue (WAT) metabolism, we determined uncoupling protein 1 (UCP1) levels in WAT. We found a main genotype effect in UCP1 (F_2,28_ = 4.995; *p* = 0.014) and interaction “sex x genotype” (F_2,28_ = 3.625; *p* = 0.0398). UCP1 protein levels were significantly elevated in Tg^5xFAD/5xFAD^ males compared to non-Tg males (*p* < 0.05), and also in Tg^5xFAD/-^ females compared to non-Tg females (*p* < 0.05) (Appendix A).

We ran a correlation test to assess if altered triglyceride and GOT levels in female 5xFAD mice were related to altered total cholesterol and cholesterol particle HDL and LDL in plasma (Table 5). The triglyceride levels were positively correlated with total cholesterol in females (*p* < 0.001 for non-Tg; *p* < 0.05 for Tg^5xFAD/-^ and Tg^5xFAD/5xFAD^ females). Triglyceride levels were also related to HDL in Tg^5xFAD/-^ females (*p* < 0.05) and Tg^5xFAD/5xFAD^ females (*p* < 0.05), and LDL in Tg^5xFAD/5xFAD^ females (*p* < 0.01). Increased GOT levels were negatively associated with LDL levels in Tg^5xFAD/5xFAD^ females (*p* < 0.05).

## 3. Discussion

AD is the most prevalent form of dementia, affecting 10% of the population over 65 years of age. Most of the approaches in neurodegenerative diseases such as AD are carried out from a cognitive perspective, focusing on structural and molecular disturbances in the hippocampus, prefrontal cortex, or motor cortex, since these brain regions are responsible for the control of memory and locomotor capacities that are the most characteristic features of AD. However, changes in body mass index (BMI) and weight loss occur in the prodromal phases and predict AD development [40,41]. Extensive research links both peripheral and brain insulin resistance with metabolic abnormalities and synaptic dysfunction, contributing to late-onset development of AD [42]. In the present study, we examined the hypothesis that early metabolic deficits observed in AD are a result of hypothalamic dysfunction and a dysregulation in metabolic hormonal signaling derived in a negative energy balance. We used both heterozygous (Tg^5xFAD/-^) and full transgenic (Tg^5xFAD/5xFAD^) mice, aiming to determine how Aβ burden could affect peripheral metabolism. Since women are at greater risk for development of AD [43], we also investigated sex differences between male and female mice. We found that 5xFAD mice had a decreased body weight, which was associated with a decreased food intake rather than changes in energy expenditure, although 5xFAD female mice exhibited a decreased energy expenditure and respiratory quotient, suggesting a sex-specific change in lipid oxidation resulting from Aβ overexpression. As revealed by glial activation, 5xFAD mice had a neuroinflammatory status in the hypothalamus despite absence of immunohistochemically traces of Aβ deposition. This phenotype was more pronounced in the full transgenic phenotype and females. Energy deficit was accompanied by low levels of plasma insulin in 5xFAD mice. Sex differences were observed as insulin signaling was defective in both male and female 5xFAD mice, but leptin signaling in the hypothalamus was impaired in females, albeit leptin levels were unaltered, as observed by secondary messengers IRS1 and STAT5, and metabolic sensors mTOR and AMPK. Intriguing results showed low orexigenic NPY levels in males but not females, which may partially explain the decreased appetite observed. All together, these observations provide evidence that hypothalamic dysfunction occurs in early–mid stages of AD, before severe neuropathology and associated cognitive deficits are evident. They also suggest that decreased food intake along with insulin impairment contribute to a negative energy balance that may explain metabolic phenotypes observed in AD patients in the prodromal stages (Figure 9).

### 3.1. Body Weight Deficit in 5xFAD Mice

Our data in the 5xFAD model of AD shows that, in the early–mid stages of cognitive decline at 6 months of age, there was an evident decrease in body weight that was more prominent in females, as both Tg^5xFAD/-^ and Tg^5xFAD/5xFAD^ showed this feature. Our results are in accordance with previous studies in 5xFAD mice of 6, 7.5, 9, and 12 months of age, showing weight loss in female full transgenic mice [44,45]. However, we also observed male weight loss in Tg^5xFAD/5xFAD^ males, but not in the Tg^5xFAD/-^ mice. Studies in different mouse models of AD such as the 3xTg have shown differences in body weight at early age, showing decreased body weight in male mice but, conversely, increased in female mice when other studies showed decreased body weight in both male and female 3xTG mice at 3 months of age [13,46]. According to previous studies in mouse models of AD, this body weight deficit seems to be mainly associated with white fat mass loss [13,43,45]. Weight loss is usually observed in AD patients, and BMI is also a marker for AD cognitive decline as changes in stages of AD are related to cumulative weight loss [47,48,49]. Moreover, it has been found that weight loss precedes one or two decades the appearance of cognitive symptoms in patients with autosomal dominant AD (ADAD) [50]. The exact causes for weight loss are, however, still under discussion and seem to be fairly related to decreased food intake and disruptions in the sleep pattern [48,49]. In our study, we found that body weight decrease was significantly associated with decreased food intake, more prominently during the active phase (night), in accordance with previous studies in the same mouse model of AD [45]. It has also been reported that behavioral factors such as anhedonia and depression in AD patients are correlated to decreased food intake and weight loss [51]. Poor food intake may lead to malnutrition in 5xFAD mice. Malnutrition is observed in AD and mild cognitive deficit (MCI) patients and correlates with cognitive performance [52]. This result supports a role of appetite in the manifestation of metabolic disturbances and a negative energy balance in AD patients and may precede cognitive decline. We cannot discard that transgenesis might direct or indirectly affect intestinal transit, nutrient absorption, and microbiota composition. Current ongoing research will address this important aspect to clarify whether weight loss has a neglected gastrointestinal component that has to be controlled to identify early metabolic dysfunctions in AD.

### 3.2. Alterations in Food Intake and Energy Expenditure in 5xFAD Mice

Along with decreased food intake, it has been proposed that hypermetabolism may also be a cause for decreased body weight in AD. There is some controversy over contribution of energy expenditure on metabolic deficits in AD patients. While some studies show increased metabolism in AD [53], others indicate decreased or non-significant changes in energy expenditure (EE) in AD patients [54]. As previously seen, the Tg4510 and 3xTg mouse models of AD show a hypermetabolic state at an early age [13,46,55]. We, in contrast, did not find increased EE when normalized per body weight and quantified by indirect calorimetry. Instead, we also observed a decreased EE in full 5xFAD transgenic mice, more preeminently in female mice during active phase (night), in accordance with the food intake pattern of activity we observed. These results show that weight loss in 5xFAD mice at 6 months of age are not related to hyperactivity nor energy expenditure, as other studies have also shown decreased motility and locomotor activity in male and female 5xFAD mice, which were associated with increased frailty [45,56]. The main differences between energy metabolism in 5xFAD and other mouse models of AD reside in either mutation leading to Tau hyperphosphorylation or behavioral traits, which may be a cause for observed hypermetabolism [55]. This seems to be consistent as suppression of transgene overexpression of Tau in the Tg4510 model of AD attenuates observed hyperactivity [57]. Surprisingly, we also found a decreased respiratory quotient (RQ) in female 5xFAD mice. This observation indicates a decline in glucose/fat fuel utilization ratio, since lower RQ measurements correlate with higher O_2_ consumption in lipid β-oxidation. This tendency in fatty acid utilization as fuel may be a triggering risk factor for the development of AD. One of the most indicative risk factors for late-onset development of AD is the presence of the APOE4 allele, causing a higher prevalence in female carriers. Previous studies have shown that mice carrying APOE4 mutation exhibit decreased body weight and fat mass, and increased fatty acid utilization as fuel [58]. A preliminary study in humans also observed that young female E4 carriers displayed lower resting EE and a redirected flux towards aerobic glycolysis instead of glucose oxidative phosphorylation. Overall, the phenomena occurring in 5xFAD mice provided supportive evidence for the fact that negative energy balance occurring in AD patients is likely related to alterations in appetite and energy utilization, and that these are drifted away from normal ranges with more intensity in females, making them more susceptible to aggravation of their metabolic dysfunctions. These data support nutritional and metabolic interventions as a useful tool for management of AD from a gender perspective.

### 3.3. Aβ Pathology and Neuroinflammation in the Hypothalamus of 5xFAD Mice

Because the hypothalamus has a marked role in integrating peripheral metabolic and hormonal signals and controlling whole-body energy homeostasis [19], we examined the Aβ pathology in the hypothalamus of 5xFAD mice in order to gain insight into the mechanisms derived in negative energy balance that we previously observed. Several studies have associated hypothalamic abnormalities in AD patients, showing decreased hypothalamus volume [59] and atrophy [60], as well as the presence of amyloid deposits and NFTs in all hypothalamic nuclei [61]. Although we did not find immunohistochemical traces of Aβ plaques in the hypothalamus of 5xFAD mice at 6 months of age, we shall not discard the possibility of influence of other forms of Aβ or APP fragments in hypothalamic neuronal status. Other mouse models of AD have stated metabolic abnormalities before detection of Aβ plaques in the brain [13]. In our case, other areas such as the hippocampus and prefrontal cortex had widespread and large Aβ plaques at 6 months of age, which can be fairly related to deficits in memory and decision tasks. There has been extensive research suggesting that soluble oligomers, which we did not observe by immunohistochemistry, can lead to activation of microglia, brain insulin resistance, and Tau hyperphosphorylation [62]. This should be highlighted as the possible cause for neuroinflammatory status observed in 5xFAD mice. This is especially relevant since glial cells are main regulators of Aβ levels in the brain and mediate Aβ clearance in the brain, and hence activation of microglia response is likely triggered by soluble Aβ [63].

Inflammation in the hypothalamus is a common cause in metabolic diseases. Neuroinflammation is observed in several hypothalamic nuclei in obesity, and cytokine infiltration worsens the disease [64]. We evaluated the expression of inflammatory markers and found an increase expression of IL-1β in full transgenic and TNF-α in heterozygous 5xFAD mice, but no changes on IL-6 expression. Although crucial information on inflammatory transducers such as NF-kB, cyclooxygenase, and nitric oxide synthase is lacking, data clearly suggest that inflammation is triggered in the hypothalamus. The exact mechanisms derived in this distinct pattern of cytokine release may depend on Aβ burden difference between heterozygous and full transgenic genotypes. However, both cytokines are part of microglia-activated response, and elevation in their expression is observed in AD patients and the hippocampus, as well as the brain cortex, in mouse models of AD. Despite the absence of Aβ plaques, Aβ oligomers may be present in the hypothalamus and bind to microglial cells, triggering a pro-inflammatory response. The immune response was accompanied by increased protein levels of GFAP (indicative of astrogliosis) and IBA1 (microgliosis). There is evidence that neuroinflammation derived from astrocytes and microglia in a context of Aβ deposition triggers Tau protein misfolding and hyperphosphorylation [65]. We observed increased Tau phosphorylation in AD led by higher levels of p25 and CDK5 in 5xFAD mice. As pointed out previously, Tau hyperphosphorylation in the hypothalamus of mice models of AD leads to increased locomotor activity and EE. Since the 5xFAD mouse does not contain Tau-specific mutations, this increased phosphorylation we observed as part of neuroinflammatory status in the hypothalamus may not be sufficient for lead and/or second hypermetabolism. The placement of the hypothalamus next to the fourth ventricle makes it highly sensitive to leakage of peripheral inflammatory molecules when disruptions on the blood–brain barrier (BBB) occur, which is also a feature of neurodegenerative processes in course with a chronic neuroinflammatory status. This inflammation of peripheral immune cells and cytokines may contribute to worsening the hypothalamic neurodegeneration.

Inflammation in the hypothalamus is thought to induce several changes in metabolism. Previous studies have also shown increased expression of IL-1β, TNF-α, and IL-6 in the hypothalamus of the 3xTg mouse model of AD [11,46]. IL-1β in the arcuate nucleus of the hypothalamus (ARC) is thought to play a major role in food intake, inducing hypophagia and reducing body weight [66]. This mechanism may account for the detrimental phenotype of the full transgenic 5xFAD mice as compared to the heterozygous 5xFAD mice in energy balance. In turn, hypothalamic TNF-α reduces thermogenesis and energy expenditure and leads to hypothalamic desensitization of insulin response [67].

### 3.4. Alterations in Plasma Hormones and Hypothalamic Signaling in 5xFAD Mice

The metabolic phenotype of 5xFAD mice was accompanied by decreased insulin, leptin, and GLP-1 hypothalamic signaling, as shown by low activity of insulin and leptin receptors and the secondary messengers IRS1 and STAT5, respectively. Moreover, low plasma levels of insulin were observed in 5xFAD mice, as well as low GLP-1 in female 5xFAD mice. It is well known that hypothalamic inflammation impairs neuronal response to neuroendocrine signals, such as insulin and leptin [68], and is observed in AD mice [69]. Both insulin are important neuroprotective growth factors, and impaired signaling of insulin, leptin, and GLP-1 in the hypothalamus are seen to contribute to worsening of Aβ pathology [8]. Recently, it has been demonstrated that cognitive impairment and AD progression are related to a defective insulin signaling in the hippocampus and frontal cortex. Postmortem analysis of human hippocampal tissue shows a correlation between high serine-inhibitory phosphorylation of IRS1 and oligomeric Aβ plaques, which were negatively associated with working memory and episodic memory [5]. The low insulin levels are likely a major contribution to the negative metabolic balance phenotype observed in 5xFAD mice. Supporting this, low-insulin plasma levels were positively correlated to body weight in 5xFAD mice. It is commonly thought that high-insulin plasma levels in pre-diabetic to type 2 diabetes mellitus (T2DM) patients contributes to AD development, with increasing evidence suggesting an exacerbation of cognitive impairment, neuroinflammation, Aβ aggregation, and tau hyperphosphorylation in AD [70,71,72]. Moreover, diabetic patients show a high prevalence and increased risk of developing AD [73]. However, low insulin levels in plasma are also a predictor for late-onset dementia and AD. Two follow-up studies have shown that low-insulin plasma levels during fasting were higher risk predictors for dementia and AD as compared to high insulin levels, and this association was independent of preclinical T2DM [74,75]. This describes a U-shaped association between fasting plasma insulin levels and risk for development of AD, as stated by studies in aging men and women [74,76]. These studies support a role of low insulin levels observed in 5xFAD mice in the early onset of AD. Other authors have shown low insulin levels in the Tg2576 mouse model of AD at 14 months of age [13]. This observation is, however, concomitant with low leptin levels in earlier age, as also seen in the 3xTg mice with weight loss [46]. Hypoleptinemia is also described in AD patients and seems to be related to low fat adipose tissue mass in weight loss and defective adipokyne section [77,78]. We also observed positive correlation between plasma leptin in male 5xFAD mice and body weight, but not in females, suggesting a difference in the mechanism of leptin secretion between both sexes. Despite the fact that we did not observe significant decreased plasma leptin levels, impaired leptin hypothalamic signaling in 5xFAD mice, which is likely related to neuroinflammation, may reproduce these mechanisms of metabolic dysfunction.

Sex bias is preeminently observed in other hormones previously described to contribute to metabolic dysfunctions aside from insulin and leptin. Along with leptin and insulin, both GIP and GLP-1 signaling in the hypothalamus are thought to contribute to insulin sensitivity and also provide a neuroprotective effect and modulate food intake and energy expenditure [79,80]. We observed that low insulin levels in male and female 5xFAD mice were related to a deficiency in both incretins. Male 5xFAD mice showed decreased basal GIP plasma levels that were also positively correlated with insulin plasma levels and body weight. Studies with GIP analogs in AD models have shown improved cognitive performance [80]. Moreover, dual GIP/GLP-1 agonists are arising as a promising therapeutic for AD as they show neuroprotective effects in AD models [81]. Specific GLP-1R stimulation reduces Aβ aggregation in vitro and in 3xTg mice in vivo. The fact that we also observed low levels of plasma GLP-1 in female mice that were positively correlated with low inulin plasma levels and body weight, but also low GLP-1R levels in the hypothalamus, suggest a worsening pattern of hypothalamic control of energy balance, especially in female mice. Ghrelin levels were also lower in female 5xFAD mice. Ghrelin also exerts a neuroprotective effect in AD independently of insulin signaling and promotes food intake [82,83]. Treatment of 5xFAD mice with ghrelin agonist MK-0677 has been shown to decrease Aβ burden, neuroinflammation, and neurodegeneration in the hippocampus [84]. Although there is no evidence of altered levels of GLP-1 and ghrelin in AD patients, the mechanisms involving both signaling pathways in the hypothalamus may account for cumulative deficits in hypothalamic control on energy balance, resulting in metabolic dysfunction. Deficiencies in neuroendocrine signaling show an early age pattern of metabolic dysfunction and weight loss.

We should highlight the finding of low levels of the adipokine resistin in the plasma of 5xFAD mice. This hormone is secreted in fatty tissue in mice and macrophages in humans and is involved in fatty acid synthesis in adipose tissue and the triggering of an inflammatory response in the brain and hypothalamic insulin resistance through its binding to TLR4 receptors [28,85,86]. Moreover, association between resistin levels and increased inflammatory markers in AD patients suggests a pro-inflammatory role of resistin in AD [87]. We found that low resistin plasma levels were positively related to low body weight in both males and females, which could also have been due to reduced fat mass in 5xFAD mice. Moreover, low hypothalamic localized resistin mRNA expression was also observed in heterozygous and full transgenic 5xFAD mice. Since central resistin infusion modulates hypothalamic neuronal activity and promotes food intake, the low levels of resistin may imply a major role in appetite regulation rather than neuroinflammatory process in 5xFAD mice. Moreover, association between resistin levels and neuroinflammatory process in the hypothalamus seems to be in the opposite direction, as we observed a negative correlation between resistin and the cytokine IL-1β in female Tg^5xFAD/-^ and Tg^5xFAD/5xFAD^ mice as well as TNF-α in female Tg^5xFAD/5xFAD^ mice. This association was also observed with body weight in female mice, suggesting an important interplay between hypothalamic neuroinflammation, resistin levels, and weight loss specifically observed in females. Our results seem to be in accordance with previous results showing a reduction in hypothalamic resistin fibers in both obese and food-deprived underweight young mice [88]. In accordance, damage induced to the ARC, where resistin co-localizes with POMC neurons, decreases resistin immunoreactivity [88]. Since metabolic dysfunction appears to be accompanied by poor endocrine signaling involving a greater range of hormones in females, this may be a cause that explains why they develop a more pronounced underweight phenotype.

### 3.5. Alterations in Energy Sensors in 5xFAD Mice

Energy sensors are constitutive of hypothalamic response to metabolites and hormones, playing a major role on hypothalamic control of energy balance. AMPK is considered the main energy sensor in the organism and its activity depends on cell energy status, having an important role on metabolic homeostasis, autophagy, cell growth, and inflammation [32]. AMPK activity is characteristic of energy requirements in the cell, and the higher phosphorylation levels observed in Tg^5xFAD/5xFAD^ mice might indicate low energy availability. The main mechanism derived from hypothalamic AMPK activation is the drive for an orexigenic response and increased appetite, implying release of NPY and AgRP mainly in the ARC [89]. The fact that we observed counter-intuitive low levels of NPY and AgRP in 5xFAD male mice shows that this response might be blunted. In fact, studies in other mouse models of AD have shown the presence of Aβ fragments 1–42 blunts response from NPY neurons [8]. Lower levels of both orexigenic and anorexigenic neuropeptides in the hypothalamus have also been associated with neuronal loss in the aged 3xTg mice of AD [11]. AMPK is also a negative regulator of the mTOR pathway, which is mainly involved in the control cell cycle, autophagy, and neuronal plasticity [90,91]. Activity of mTOR is also defined by energy requirements of the cell and is a key element of the insulin signaling pathway, deriving an anabolic response. It has been defined that brain insulin resistance occurs primarily with early hyperactivation of insulin signaling, including mTOR hyperactivity, which causes negative feedback on the insulin receptor, IRS1 and mTOR itself [6]. The fact that we observed low mTOR activity in 5xFAD mice measured by its phosphorylation on the residue Ser2448 is in accordance with low insulin levels and decreased pTyr/pSer ratio on IRS1, which translates into decreased insulin responsiveness in the hypothalamus. mTOR opposes AMPK-mediated autophagy. Hence, low mTOR activity and increased autophagy may help Aβ clearance, as previously seen [92]. However, decreased mTOR activity has been observed in 5xFAD mice, APP/PS1 AD mice, and AD patients, being impaired as a result of exposure to Aβ peptides, contributing to synaptic and cognitive deficiencies [93,94,95]. Moreover, low mTOR signaling has been related to increased cytokine expression in AD rats [96]. Hence, both impaired response to hormonal cues and enhanced cytotoxic activity of Aβ may exacerbate hypothalamic insulin signaling response and metabolic sensing through AMPK and mTOR, contributing to the negative energy balance observed in 5xFAD mice. Along with AMPK and mTOR, ERK’s role in the hypothalamic control of energy homeostasis is fairly involved in growth factor signaling as a second messenger in the Ras–ERK pathway, with some events of crosstalk with the mTOR pathway as an anabolic response [33]. However, we observed no influence of total ERK activity as measured by its phosphorylation, but total protein levels tended to be decreased in males as opposed to higher levels in females. We also observed higher levels of total mTOR in both Tg^5xFAD/5xFAD^ males and females. Higher total mTOR levels and lower mTOR relative activity as measured by p-mTOR/mTOR ratio has been observed in normal aging process in the hippocampi of mice [6]. Because insulin signaling in the hypothalamus was decreased, we also evaluated and observed low hypothalamic GSK3-β activity in 5xFAD mice (Appendix A). Both mTOR and GSK3-β have been previously related to Tau hyperphosphorylation [97]. However, our results suggesting increased Tau phosphorylation in 5xFAD mice seem to be related to CDK5 and neuroinflammation [98].

### 3.6. Alterations in Neuropeptides in 5xFAD Mice

Ultimately, hypothalamic control of food intake resides on the release of orexigenic/anorexigenic neuropeptides. NPY/AgRP are co-localized in neurons in the ARC, promoting food intake and lowering energy expenditure, whereas POMC/CART neurons do the opposite [19,35,36,37]. Previous studies based on other mouse models of AD with hypoleptinemia assumed hypothalamic neurons had abnormal or absent responses to hormonal cues, causing the metabolic deficits observed [13,99]. In our study, we confirmed that the hypothalamus of the 5xFAD mouse model of AD showed a blunted response to insulin and leptin in both male and female mice, as well as low plasma levels of ghrelin, GLP-1, and resistin in female mice, promoting an exacerbated status of negative energy balance. The heterozygous model resulted in a less pronounced phenotype, especially in male mice. However, distinct patterns of food intake and activity were observed in accordance with hypothalamic neuropeptide expression. We found decreased NPY and AgRP levels in heterozygous and full transgenic 5xFAD male mice after a night of food deprivation. This result agrees with the decreased food intake pattern in the full transgenic Tg^5xFAD/5xFAD^ mice. However, lower expression of orexigenic neuropeptides in the heterozygous Tg^5xFAD/-^ mice did not account for a significant decrease in food intake. Nevertheless, we did find a peak in EE in the last hours of nighttime (active phase) and first hours of daytime (resting phase) in heterozygous Tg^5xFAD/-^ mice. Heterozygous mice showed a tendency towards increased intake during the day, which, in the case of males, was likely related to increased expression of orexin in the hypothalamus. Orexin promotes wakefulness, appetite, and EE, and its expression is inhibited by NPY projections [100]. This is in accordance with the negative correlation we observed between NPY and orexin expression in the hypothalamus. The fact that mice were sacrificed at the very first hours of daytime, when energy expenditure was slightly increased in heterozygous mice, might imply why orexin levels were higher. Both male and female 5xFAD mice have shown decreased sleep at 4–6.5 months of age, and sleep disruption was more prominent during late hours of night time and early hours of day time, as matched with our results of EE in full transgenic mice and more pronounced in heterozygous 5xFAD mice at the same age [101]. The reason why we observed this pattern in a more pronounced way in heterozygous mice may rely upon a possible inability of orexin to compensate for low NPY levels in full transgenic Tg^5xFAD/5xFAD^ mice.

As previously reported in the Tg2576 mouse model of AD, hypothalamic NPY neurons lose response to leptin, which has a hyper-polarizing effect, and ghrelin, which in turn depolarizes NPY neurons and promotes neuropeptide release. This effect is mediated by the presence of Aβ peptides [13]. The fact that we did not observe changes in NPY/AgRP mRNA expression in female mice could have been a result of small fasting time before sacrifice. Changes in NPY expression in the hypothalamus of Tg2576 mice were observed after 48h of food deprivation, but not in the fed state [13]. However, male Tg2576 mice showed low POMC and CART expression without food deprivation, which resembles the non-significant tendency we observed in POMC mRNA expression in 5xFAD female mice after a short period of food deprivation. In our study, NPY and POMC expressions were positively correlated in female 5xFAD mice, as it would be expected to be inversely correlated. This observation could imply a mechanism of generalized neuronal dysfunction in the hypothalamus, leading to impaired food intake or energy expenditure response during fasting.

### 3.7. Plasma Lipid Profile and Increased PAI-1 Levels in Female 5xFAD Mice

PAI-1 is an inhibitor of tissue (tPA) and urokinase plasminogen activator (uPA), playing an important role in inhibiting fibrinolysis, hence decreasing blood clot clearance. Because of this, increased levels of plasma PAI-1 are observed in obesity and metabolic syndrome, leading to increased risk of atherosclerosis development. Our results showed a sex-biased increment in plasma PAI-1 levels in female Tg^5xFAD/-^ and Tg^5xFAD/5xFAD^ mice and a negative correlation between body weight and PAI-1 plasma levels in the aforementioned groups. Interestingly, altered PAI-1 levels have been observed in the brain of APP/PS1 mice and AD patients, being increased in correlation to age and progress of dementia. Studies in mice have showed inhibition of PAI-1 [102]. Addition of PAI-1 inhibitors in APP/PS1 and Tg2576 AD mice improve clearance and reduce levels of plasma and brain Aβ and restore memory function [103,104]. The fact that plasma levels of PAI-1 are altered and seem to relate to prodromal weight loss suggest that systemic metabolic imbalance may be related to worsening of AD pathology in females in a mechanism different from obesity-related increments of PAI-1 levels.

We investigated plasma lipid profile of 5xFAD AD mice because several studies relate altered plasma lipid levels in AD patients. We observed again a sex-specific decay in triglycerides and total cholesterol of female 5xFAD mice that were positively correlated. From a metabolic point of view, depletion of triglycerides in female 5xFAD mice may be indicative of a shift in lipid utilization as energy source, in accordance with the results observed and previously discussed on the low RQ observed and increased lipid utilization. In fact, increased expression of WAT UCP1 and UCP2 was also observed in Tg^5xFAD/-^ and Tg^5xFAD/5xFAD^ female mice. Previously, it has been observed patients with probable AD have abnormally low levels of total cholesterol, triglycerides, and LDL cholesterol when compared to control groups, and these low levels were inversely but not significantly correlated with cognitive performance. The specific reason why blood lipids are altered in AD patients is not fully understood. Normally, higher levels of total cholesterol, triglycerides, and LDL are related to metabolic syndrome and increased risk of AD, which implies a specific relationship between obesity-derived problems and AD. Some other studies have assessed increased risk of dementia in patients with higher mid-life and late-life levels of total cholesterol [38,39]. However, a drastic decrease in total cholesterol occurs before dementia diagnosis and is predictive of dementia appearance [39]. Cholesterol is a key element for several processes such as maintenance and fluidity of cell membrane, synaptic transmission, and synthesis of steroid hormones, all of which may experience a negative outcome in AD patients. It should be noted that full transgenic Tg^5xFAD/5xFAD^ female mice had significantly low LDL and high HDL levels in plasma. It has been shown that non-carriers of APOE4 allele AD patients manifest an improvement in cognitive performance with lipid therapy increasing LDL levels, whereas an increment in HDL levels were associated with a poorer score in cognitive tests [105]. Our results indicate that mechanisms of dysregulation of lipid metabolism and altered lipid profile may contribute to worsening of AD pathology, specifically in the higher incidence in females as compared to males in a different pattern non-related to AD prevalence in obesity and diabetic patients.

### 3.8. Limitations of the Study

Some differences between AD patients and the 5XFAD mouse model of AD must be taken into consideration when pointing out the results hereby obtained. 5xFAD mice contain five mutations, two of the in presenilin-1 (PSEN-1) and three in APP, leading to early production of Aβ fragments. 5xFAD mice hence develop early cognitive impairments similar to those of AD patients. However, 5xFAD mice are unable to develop NFTs and do not develop severe Tau hyperphosphorylation [106]. Although we observe increased Tau phosphorylation in the hypothalamus of 5xFAD mice, this does not seem to reproduce the hypermetabolic status observed in AD mice bearing mutations in Tau protein such as 3xTg [12,13]. However, the higher similarities in neurodegeneration and cognitive impairment in the 5xFAD with AD patients with respect to other AD mice may imply Aβ pathology linked to metabolic dysfunction could be reproduced with more accuracy in our mouse model of AD. As our mice were fed standard chow diet, we could not ascertain whether a high-fat diet (HFD) (a very common nutritional problem in western countries) has or does not have an impact in the progression of metabolic and cognitive decline in AD. Since 5xFAD mice presented impairments in insulin, leptin, and resistin signaling, this could lead to a hyper-sensitive and detrimental response to exposure to fat-enriched nutrition, as this has been previously observed in other mouse models of AD [46]. A previous study has shown detrimental effects of HFD in 5xFAD glucose tolerance, lipid profile, and microbiome composition [44]. However, we are still lacking information as to whether HFD could differently affect the heterozygous non-full transgenic bearing less Aβ burden and also if 5xFAD mice, which have been proven to be more susceptible to metabolic impairments, could exert a more deteriorated phenotype. Moreover, the results observed did not focus on the neuroendocrine system and hypothalamic regulation of energy balance. Future studies should address these questions thoroughly and comparisons should be made between 5xFAD and other AD mice.

In addition, the expression of either Aβ oligomers or plaque deposition in peripheral tissues (including gut mucosa and myenteric plexus, autonomic ganglia, pancreatic islets, liver, and adipose tissue) has to be addressed in order of clarify the contribution of peripheral tissues when amyloid deposition is boosted by mutations.

## 4. Conclusions

In conclusion, we found a negative energy balance in 5xFAD mice leading to weight loss at 6 months of age that was related to alterations in food intake, hypothalamic dysfunction despite absence of observable Aβ plaques, and neuroendocrine dysregulation. Decreases in body weight were differently regulated and more severe in female 5xFAD mice and worsened with Aβ burden in the full transgenic genotype. The existence of clear differences regarding the heterozygous versus homozygous conditions is remarkable, a fact that might help in the search for models that boost amyloid deposition where there is not a complete penetration mutation. Thus, heterozygous animals have to be included in future studies as a way of analyzing non-genetic AD contributing factors.

In addition, we observed low plasma levels of insulin and insulin-releasing hormones in both male and female 5xFAD mice that were related to decreased body weight and decreased hypothalamic insulin and leptin signaling. Neuroinflammation and low resistin levels in the hypothalamus were specifically related to weight loss in female mice, which also exhibited altered plasma lipid profile, likely worsening the whole-body energy homeostasis. This study provides useful information for detection of possible early metabolic markers indicative of the outcome of AD and serve as possible therapeutic targets in a sex-specific point of view. This strategy should help in stratification of AD patients. Our work also points out a different focus on ageing-related prodromal weight loss as a comorbid state of AD and focus on less-explored areas in the brain AD pathology such as the hypothalamus, opening the scope of further research in AD.

## 5. Materials and Methods

### 5.1. Ethics Statement

The research procedures were approved by the Research and Clinical Ethics Committee of the Regional University Hospital of Malaga and the University of Malaga. All experimental procedures with animals were carried out in strict accordance with the guidelines of Royal Decree 1201/2005 of 21 October 2005 (BOE no. 252), and in compliance with Directive 86/609/ECC of the European Community (24 November 1986) in relation to the regulation of research with animals. All efforts were made to minimize the suffering of the animals, as well as to reduce the number of animals used.

### 5.2. Animals

Animals used were 6-month-old male and female non-transgenic (non-Tg) mice and heterozygous (Tg^5xFAD/-^) and homozygous (Tg^5xFAD/5xFAD^) transgenic 5xFAD mice. 5xFAD mice co-express and co-inherit familial Alzheimer disease (FAD) mutant forms of human APP (the Swedish mutation: K670N, M671L; the Florida mutation: 1716V; the London mutation: V7171) and PSI (M146L; L286V) transgenes under transcriptional control of the neuron-specific mouse Thy-1 promotor (Tg6799 line) [15]. 5XFAD lines (B6/SJL genetic background) were maintained by crossing heterozygous transgenic mice with B6/SJL F1 breeders (The Jackson Laboratory, Bar Harbor, ME, USA). Non-Tg wild-type littermate mice served as a control.

The rodents were housed in the Animal Center for Experimentation at the University of Malaga. This center complies with all current regulations for breeding and housing. The animals were housed individually with free access to food and water under standardized conditions: 20 ± 2 °C of room temperature, relative humidity of 40 ± 5%, and light/dark cycle of 12 h with dawn/sunset effect. The mice were fed on a standard pellet diet (STD) (3.02 Kcal/g with 30 Kcal% protein, 55 Kcal% carbohydrate, and 15 Kcal% fat; purchased from Harlam (Tecklad, Madison WI, USA)). All study animals were sacrificed at 6 months of age with sodium pentobarbital (150 mg/kg, i.p.). Blood was drawn directly from the right atrium and perfused with 0.1M PBS.

### 5.3. Sample Collection

Blood was drawn directly from the right atrium and perfused with 0.1M PBS. Blood was centrifuged (2100× *g* for 10 min, 4 °C), and the plasma was kept at −80 °C for a biochemical analysis. Histology samples were obtained from the left hemisphere of the brain and kept in 4% paraformaldehyde for 48 h. Samples for biochemical analysis were also obtained from the right hemisphere of the brain and brown fat. Samples from right hemisphere and brown fat were flash frozen in liquid nitrogen, then stored at −80 °C until analysis.

### 5.4. Measurement of Metabolites in Plasma

The following plasma metabolites were measured: glucose, urea, creatinine, triglycerides, total cholesterol, HDL cholesterol, LDL cholesterol and the hepatic enzymes glutamic oxaloacetic transaminase (GOT) and glutamate pyruvate transaminase (GPT). These metabolites were analyzed using commercial kits, according to the manufacturer’s instructions, and a Hitachi 737 Automatic Analyzer (Hitachi Ltd., Tokyo, Japan). Each metabolite concentration was expressed in mg/dL. Hepatic enzyme levels were expressed in IU/L.

### 5.5. Bio-Plex Pro Multiplex Diabetes Assay

Plasma levels of hormones insulin, glucagon, ghrelin, leptin, glucagon-like peptide-1 (GLP-1), plasminogen activator inhibitor-1 (PAI-1), gastric inhibitory polypeptide (GIP), and resistin were determined by multiplex immunoassay system using commercial kits: Bio-Plex Pro™ mouse diabetes 8-plex immunoassay (Bio-Rad, Hercules, CA, USA, cat. number: #171F7001M). Plates were run on a Bio-Plex MAGPIX™ Multiplex Reader with Bio-Plex anager™ MP Software (Luminex, Austin, TX, USA). Hormone concentrations were expressed in pg/mL, and detection limits were 68.29 (insulin), 0.50 (glucagon), 0.64 (ghrelin), 5.07 (leptin), 0.59 (GLP-1), 2.98 (PAI-1), 4.31 (GIP), and 184.89 (resistin) pg/mL.

### 5.6. RNA Isolation and RT-qPCR Analysis

We performed real-time PCR (TaqMan, ThermoFisher Scientific, Waltham, MA, USA) as described previously [107] using specific sets of primer probes from TaqMan^®^ Gene Expression Assays, as shown in Appendix A. Total RNA was extracted from tibia samples using the Trizol^®^ method according to the manufacturer’s instructions (ThermoFisher Scientific). RNA samples were isolated with RNAeasy minelute cleanup-kit including digestion with DNase I column (Qiagen) and quantified using a spectrophotometer to ensure A260/280 ratios of 1.8–2.0. After the reverse transcript reaction from 1 μg of mRNA, a quantitative real-time reverse transcription polymerase chain reaction (qPCR) was performed in a CFX96TM Real-Time PCR Detection System (Bio-Rad, Hercules, CA, USA) and the FAM dye labeled format for the TaqMan^®^ Gene Expression Assays (ThermoFisher Scientific). A melting curve analysis was performed to ensure that only a single product was amplified. After analyzing several reference genes, we normalized values obtained from the tibia samples in relation to Actb levels (Mm02619580_g1, amplicon length: 143; ThermoFisher Scientific), which were found not to vary significantly between experimental groups.

### 5.7. Protein Extraction and Western Blot Analysis

Total protein from 5–15 mg of hypothalamic samples was extracted using 500 μL ice-cold cell lysis buffer for 30 min, as previously described [108,109]. A quantity of 50 μg of protein was resolved on a 4–12% (Bis-Tris) Criterion XT Precast Gel (Bio-Rad Laboratories, Inc., Hercules, CA, USA, cat. number: 3450124), and then transferred onto nitrocellulose membranes (Bio-Rad Laboratories, Inc., Hercules, CA, USA). Total protein content was visualized after staining with Ponceau red. Membranes were blocked in TBS-T (50 mM Tris-HCl (pH 7.6), 200 mM NaCl, and 0.1% Tween 20) with 2% albumin fraction V from BSA (Roche, Mannheim, Germany) for 1h at room temperature. For specific protein detection, the membrane was incubated overnight at 4 °C in TBS-T containing 2% BSA and the corresponding primary antibody (Appendix A). Mouse γ-adaptin was used as the reference protein. After several washes in TBS-T containing 1% Tween 20, an HRP-conjugated anti-rabbit or anti-mouse IgG (H+L) secondary antibody (Promega, Madison, WI, USA) diluted 1:10,000 was added, followed by incubation for 1 h at room temperature. After extensive washing in TBS-T, the membranes were incubated for 1 min with the Western Blotting Luminol Reagent kit (Santa Cruz Biotechnology, Santa Cruz, CA, USA), and the specific protein bands were visualized and quantified by chemiluminescence using a Chemi-Doc TM MP Imaging System (Bio-Rad, Barcelona, Spain). For specific detection of phosphorylated form of proteins, after measuring phosphorylation proteins, the specific antibodies were removed from membrane by incubation with stripping buffer (2% SDS, 62.5 mM Tris HCl (pH 6.8), 0.8% β-mercaptoethanol) for 30 min at 50 °C. Membranes were extensively washed in ultrapure water, and then re-incubated with the corresponding antibody specific for the total protein. Quantification of results was performed using ImageJ software (http://imagej.nih.gov/ij, accessed on 5 May 2020). The results are expressed as either the phosphorylated form of target protein/target protein ratios or target protein/γ-adaptin ratios. Results for Non-Tg male protein levels were arbitrarily set as 1.

### 5.8. Immunohistochemistry

Left hemisphere of the brains were post-fixed in 4% paraformaldehyde for 48h and cryopreserved in 30% sucrose in 0.1M PBS solution for at 4 °C until processing. Free-floating coronal sections of mouse hypothalamus and hippocampus were selected from −1.22 to −1.94 mm of Bregma levels [110]. Serial sections were blocked with 5% donkey serum, 0.5% Triton X-100 in 0.1M PBS for 45 min at room temperature, as previously described [111]. For plaque amyloid-β analysis, we used rabbit anti-Aβ (1:500, Abcam). For neuroinflammation analysis, we used rabbit anti-glial fibrillary acidic protein (GFAP; 1:1000; Dako) and rabbit anti-Iba1 (1:500, Abcam). Primary antibodies were incubated overnight at room temperature. After rinsing, the sections were incubated with secondary antibody biotinylated goat anti-rabbit (1:500, GE Healthcare) for 2 h at room temperature. All antibodies were diluted in PBS, 0.5% Triton X-100, and 2.5% donkey serum (Sigma-Aldrich, St. Louis, MO, USA). We used the peroxidase-conjugated ExtraAvidin method and diaminobenzidine as the chromogen to visualize the reaction product. Quantification was performed using ImageJ software (http://imagej.nih.gov/ij, accessed on 5 May 2020). Total amyloid plaques were counted using three binarized sections of hypothalamus and hippocampus per animal.

### 5.9. Measurement of Energy Expenditure and Respiratory Quotient

For 48 h, mice were analyzed for energy expenditure (EE, kcal/kg lean mass), respiratory quotient (RQ, VCO_2_/VO_2_), and food intake using a calorimetric system (LabMaster, TSE System, Bad Homburg, Germany) as previously described [112]. This system is an open-circuit instrument that determines (1) the energy consumed by the amount of caloric intake (kilocalories) along time (hours) and normalized by the lean mass (kilograms) and (2) the ratio between the CO_2_ production and O_2_ consumption (VCO_2_/VO_2_). Activity was measured by infrared system counting mice movement and rearing. Previously, all rats were acclimated to the experimental room and habituated to the system for 48 h before starting the measurements.

### 5.10. Statistical Analysis

Graph-Pad Prism 7.0 software was used to analyze the data. Values are represented as mean ± standard error of the mean (SEM) for each in vivo experimental group, according to the assay. The significance of differences within and between groups was evaluated by a two-way analysis of variance (ANOVA), factors: “genotype” x “sex”, followed by Tukey post-hoc test for multiple comparisons. Alternatively, for comparisons between two groups, Student *t*-test was also used. A *p*-value ≤ 0.05 was considered to be statistically significant. (* = *p* < 0.05; ** = *p* < 0.01; *** = *p* < 0.001) versus same-sex non-Tg group. (# = *p* < 0.05; ## = *p* < 0.01; ### = *p* < 0.001) versus same-sex Tg^5xFAD/-^ group.

## Figures and Tables

**Figure 1 ijms-22-05365-f001:**
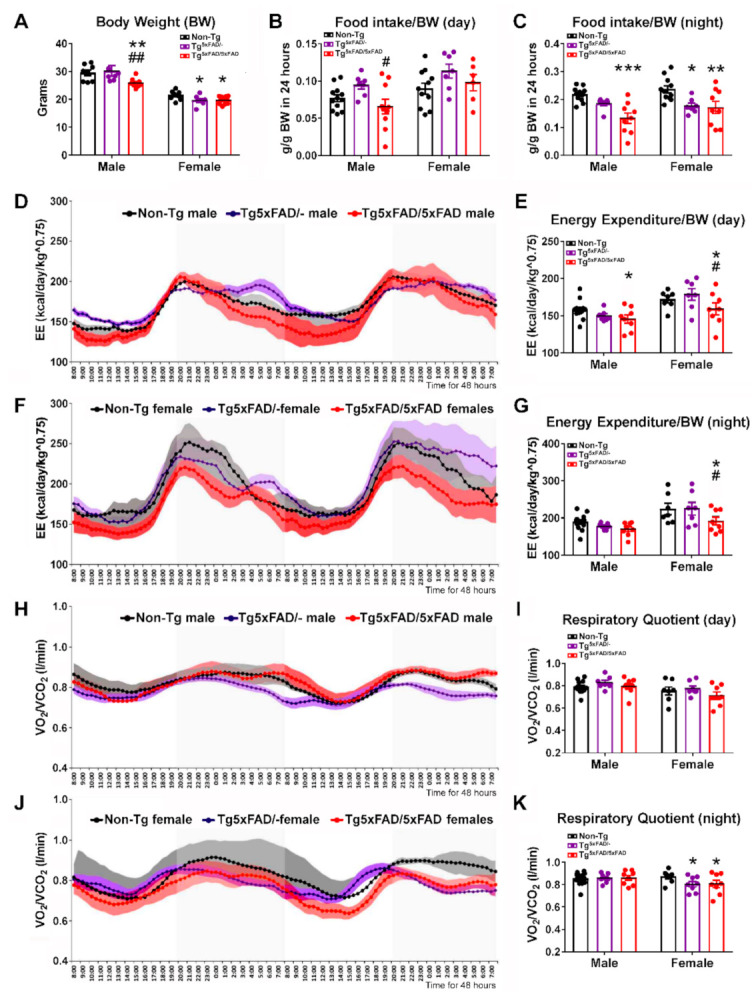
Negative energy balance in 5xFAD mice was associated with decreased food intake and energy expenditure for 48 h. (**A**) Body weight showing significantly decreased body weight in Tg^5xFAD/5xFAD^ males, Tg^5xFAD/-^ females, and Tg^5xFAD/5xFAD^ females at 6 months of age. (**B**,**C**) Decreased food intake per body weight in Tg^5xFAD/5xFAD^ males during the day (light phase, 8 a.m. to 8 p.m.) and night (dark phase, 8 p.m. to 8 a.m.). Decreased food intake also occurred in Tg^5xFAD/-^ females and Tg^5xFAD/5xFAD^ females during the night. (**D**–**G**) Energy expenditure normalized per body weight (EE/BW) in males and females, showing decreased mean EE/BW in Tg^5xFAD/5xFAD^ males during the day, and in g^5xFAD/5xFAD^ females during the day and night. (**H**–**K**) Respiratory quotient showing decreased ratio of vO2/vCO2 in Tg^5xFAD/-^ females and Tg^5xFAD/5xFAD^ females during the night, indicating decreased glucose utilization/increased fatty acid oxidation as energy source. *n* = 7–15 per group. Two-way ANOVA analysis with Tukey’s post hoc test: * = *p* < 0.05, ** = *p* < 0.01, *** = *p* < 0.001 versus same-sex non-Tg group; # = *p* < 0.05, ## = *p* < 0.01 versus same-sex Tg^5xFAD/-^ group.

**Figure 2 ijms-22-05365-f002:**
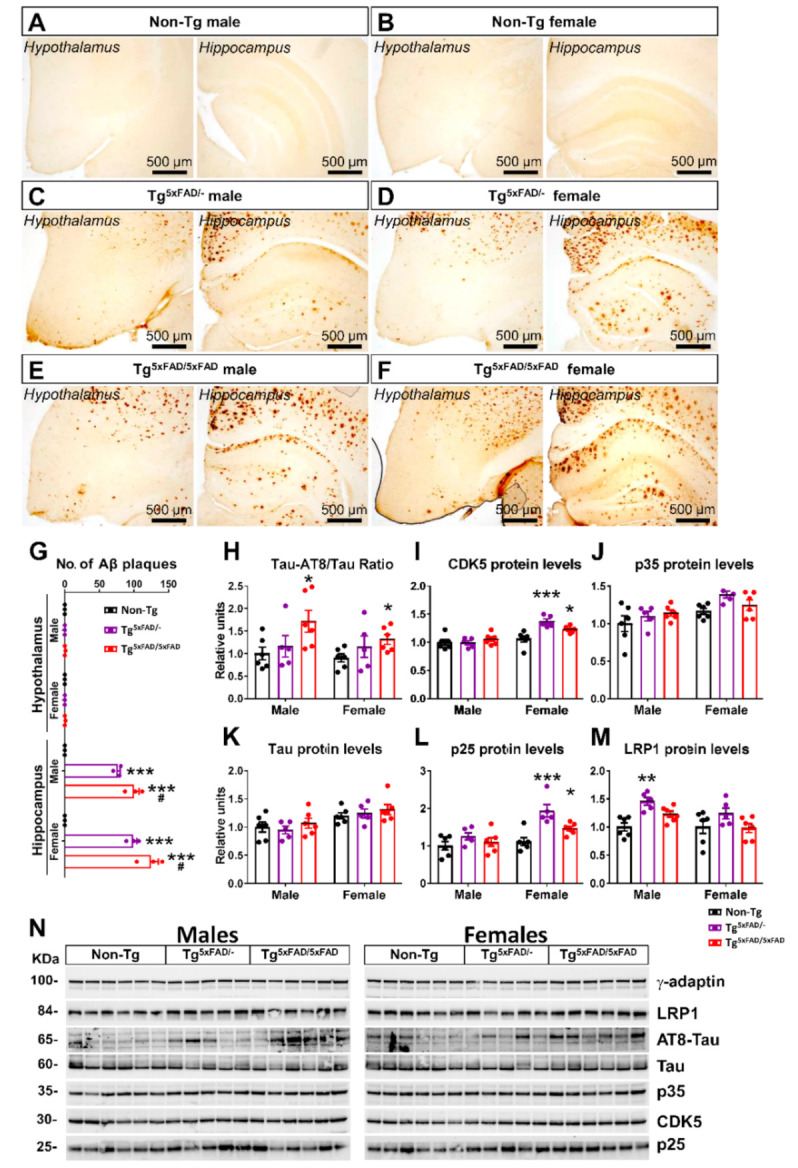
Absence of amyloid-β 42 fragments in hypothalamus of 5xFAD mice at 6 months of age but increased hypothalamic Tau phosphorylation. (**A**–**F**) Immunohistochemical sections of Aβ_42_ in the hypothalamus and hippocampus of 5xFAD mice, showing the presence of Aβ_42_ plaques in hippocampus of Tg^5xFAD/-^ and Tg^5xFAD/-^ males and females, but in the absence in the hypothalamus in all the groups. (**G**) Quantification of total Aβ plaques in the hypothalamus and hippocampus of 5xFAD mice, showing the absence of Aβ in hypothalamic sections and higher total Aβ content in both male and female Tg^5xFAD/5xFAD^ mice with respect to Tg^5xFAD/+^ mice. (**H**–**M**) Densitometric evaluation of Tau activation (Tau-AT8/Tau), CDK5 protein levels, p35 protein levels, Tau protein levels, p25 protein levels, and LRP1 protein levels (*n* = 5–6 per group). Protein levels were normalized with γ-adaptin. Associated phosphorylations were normalized with respective total protein levels. Non-Tg males were set as 1 for protein relative units. (**N**): representative Western blot images. Two-way ANOVA analysis with Tukey’s post hoc test: * = *p* < 0.05, ** = *p* < 0.01, *** = *p* < 0.001 versus same-sex non-Tg group; # = *p* < 0.05 versus same-sex Tg^5xFAD/-^ group.

**Figure 3 ijms-22-05365-f003:**
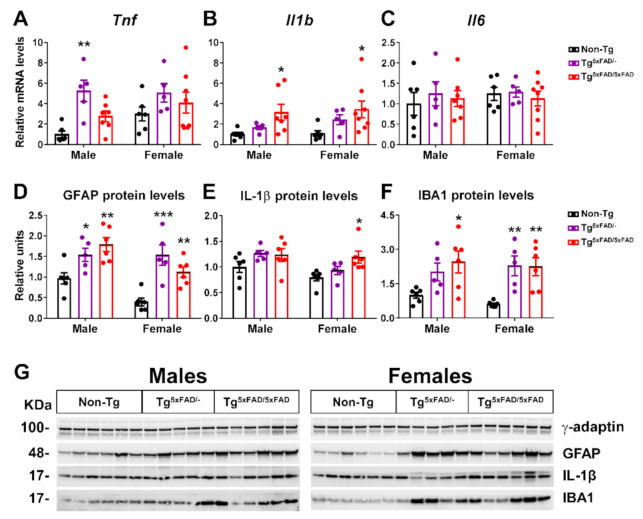
Inflammatory markers were present in the hypothalamus of 5xFAD mice at 6 months of age. (**A**–**C**) Representative quantification of expression of cytokine genes (TNF-α, IL-1β, IL-6) in hypothalamus assessed by qPCR (*n* = 5–8 per group). (**D**–**F**) Representation of Western blot membranes and (**G**) densitometric evaluation of GFAP, IL-1β, and IBA1 protein levels (*n* = 5–6 per group). Protein levels were normalized with γ-adaptin. Non-Tg males were set as 1 for relative mRNA units and protein relative units. Two-way ANOVA analysis with Tukey’s post hoc test: * = *p* < 0.05, ** = *p* < 0.01, *** = *p* < 0.001 versus same-sex non-Tg group.

**Figure 4 ijms-22-05365-f004:**
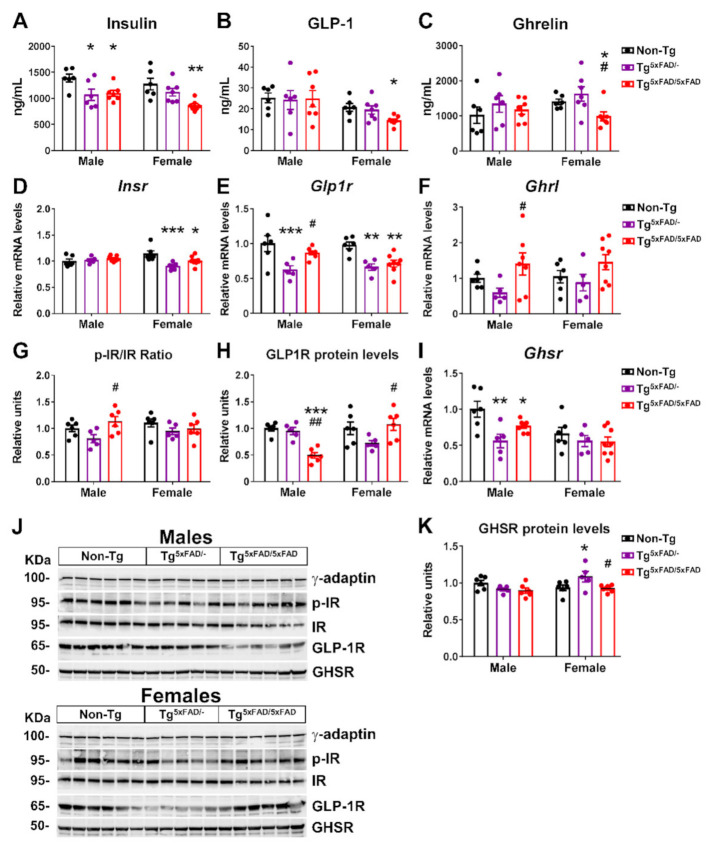
Altered insulin, leptin, and GLP-1 plasma levels and hypothalamic receptors were genotype and sex-specific in 5xFAD mice at 6 months of age. (**A**–**C**) Plasma levels of insulin, leptin, and GLP-1 (*n* = 6–7 per group). (**D**–**F**) Representative quantification of expression of insulin receptor (*Insr*), leptin receptor (*Lepr*), and GLP-1 receptor (*Glp1r*) in hypothalamus assessed by qPCR (*n* = 5–8 per group). (**G**–**I**) Representation of Western blot membranes, and (**J**) densitometric evaluation of insulin receptor (IR) activation (p-IR/IR), leptin receptor (LepR) activation (p-LepR/LepR), and GLP-1 receptor (GLP1R) protein levels (*n* = 5–6 per group). Protein levels were normalized with γ-adaptin. Associated phosphorylations were normalized with respective total protein levels. Non-Tg males were set as 1 for relative mRNA units and protein relative units. Two-way ANOVA analysis with Tukey’s post hoc test: * = *p* < 0.05, ** = *p* < 0.01, *** = *p* < 0.001 versus same-sex non-Tg group; # = *p* < 0.05, ## = *p* < 0.01 versus same-sex Tg^5xFAD/-^ group.

**Figure 5 ijms-22-05365-f005:**
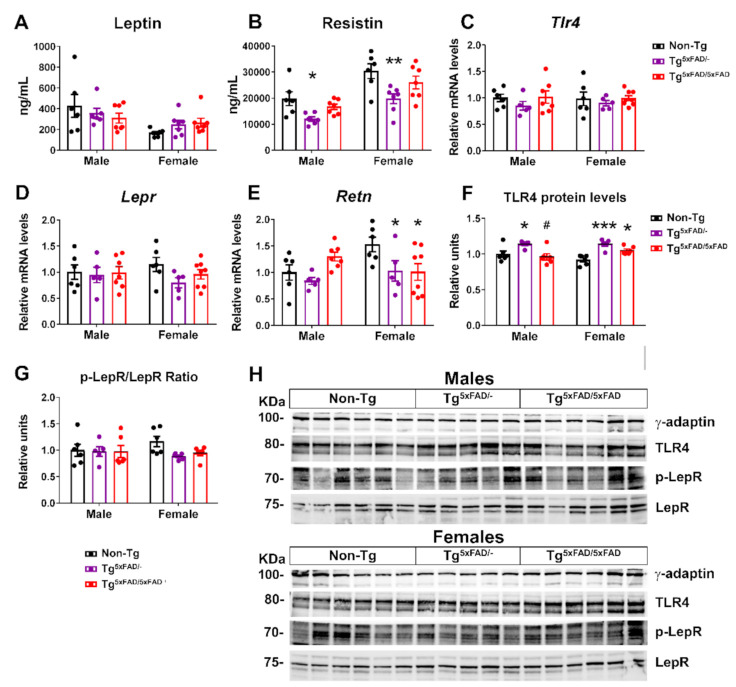
Decreased ghrelin and resistin plasma levels, with hypothalamic receptors being more specific in 5xFAD female mice at 6 months of age. (**A**,**B**) Plasma levels of ghrelin and resistin (*n* = 6–7 per group). (**C**–**F**) Representative quantification of expression of hypothalamic ghrelin (*Ghrl*), resistin (*Retn*), ghrelin receptor (*Ghsr*), and resistin putative receptor (toll-like receptor 4, *Tlr4*) assessed by qPCR (*n* = 5–8 per group). (**G**–**H**) Representation of Western blot membranes and (**I**) densitometric evaluation of ghrelin receptor (GHSR) and resistin putative receptor (TLR4) protein levels (*n* = 5–6 per group). Protein levels were normalized with γ-adaptin. Non-Tg males were set as 1 for relative mRNA units and protein relative units. Two-way ANOVA analysis with Tukey’s post hoc test: * = *p* < 0.05, ** = *p* < 0.01, *** = *p* < 0.001 versus same-sex non-Tg group; # = *p* < 0.05 versus same-sex Tg^5xFAD/-^ group.

**Figure 6 ijms-22-05365-f006:**
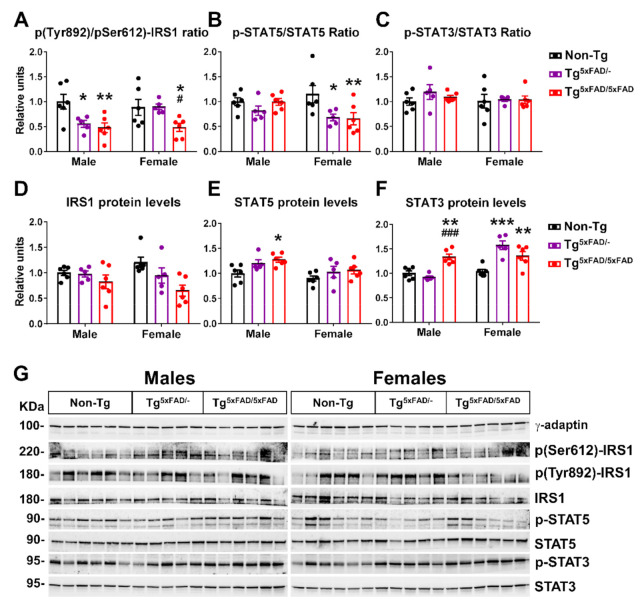
Insulin and leptin hypothalamic signaling were decreased in 5xFAD mice at 6 months of age, as shown by secondary messengers IRS1 and STAT5. (**A**–**F**) Representation of Western blot membranes and (**G**) densitometric evaluation of insulin receptor substrate 1 (IRS1) activation ratio (pTyr692-IRS1/pSer612-IRS1), IRS1 protein levels, signal transducer and activator of transcription 5 (STAT5) activation (p-STAT5/STAT5), STAT5 protein levels, STAT3 activation (p-STAT3/STAT3), and STAT3 protein levels (*n* = 5–6 per group). Protein levels were normalized with γ-adaptin. Associated phosphorylations were normalized per activation/inhibition phosphorylation ratio (IRS1) or respective total protein levels (STAT5 and STAT3). Non-Tg males were set as 1 for protein relative units. Two-way ANOVA analysis with Tukey’s post hoc test: * = *p* < 0.05, ** = *p* < 0.01, *** = *p* < 0.001 versus same-sex non-Tg group; # = *p* < 0.05, ### = *p* < 0.001 versus same-sex Tg^5xFAD/-^ group.

**Figure 7 ijms-22-05365-f007:**
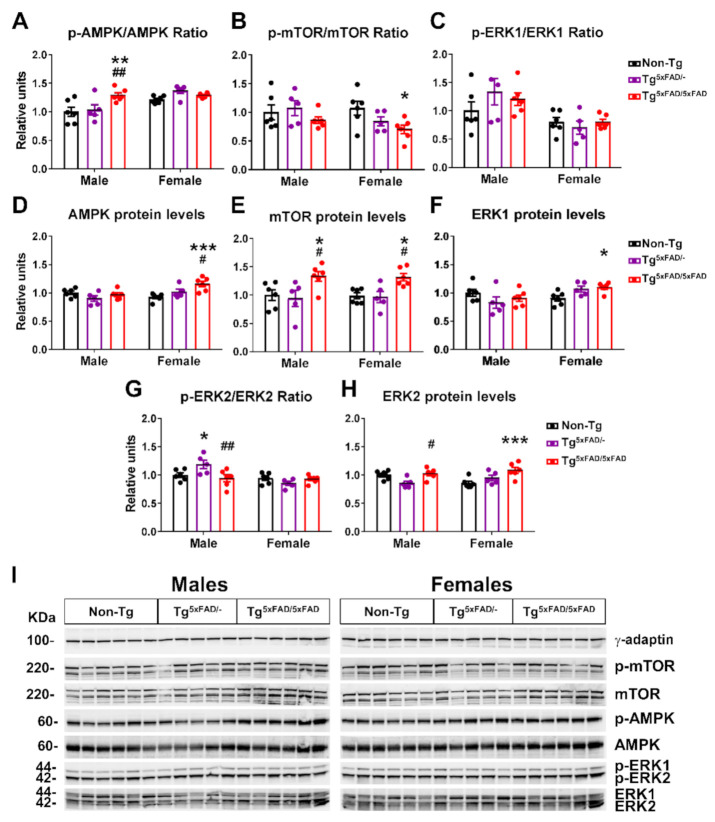
Metabolic sensors AMPK, mTOR, and ERK were altered as a consequence of impaired insulin signaling in the hypothalamus of 5xFAD mice at 6 months of age. (**A**–**H**) Representation of Western blot membranes and (**I**) densitometric evaluation of AMP kinase (AMPK) activation (p-AMPK/AMPK), AMPK protein levels, mammalian target of rapamycin (mTOR) activation (p-mTOR/mTOR), extracellular-regulated kinase 1 (ERK1) activation (p-ERK1/ERK1), ERK1 protein levels, ERK2 activation (p-ERK2/ERK2), and ERK2 protein levels (*n* = 5–6 per group). Protein levels were normalized with γ-adaptin. Associated phosphorylations were normalized with respective total protein levels. Non-Tg males were set as 1 for protein relative units. Two-way ANOVA analysis with Tukey’s post hoc test: * = *p* < 0.05, ** = *p* < 0.01, *** = *p* < 0.001 versus same-sex non-Tg group; # = *p* < 0.05, ## = *p* < 0.01 versus same-sex Tg^5xFAD/-^ group.

**Figure 8 ijms-22-05365-f008:**
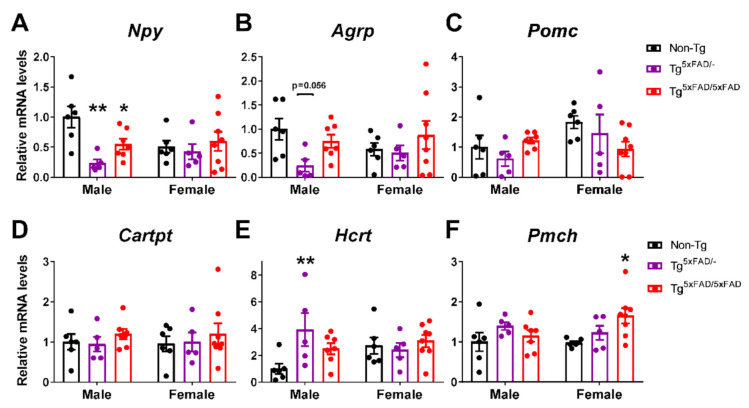
Hypothalamic orexigenic neuropeptide expression was decreased in 5xFAD males accompanied by altered orexin expression in males and MCH expression in females. (**A**–**F**) Representative quantification of expression of hypothalamic neuropeptide Y (*Npy*), agouti-related peptide (*Agrp*), pro-opiomelanocortin (*Pomc*), cocaine and amphetamine-regulated transcript prepropeptide (*Cartpt*), orexin (hipocretin; *Hcrt*), and ppre-melanin concentrating hormone (*Pmch*) assessed by qPCR (*n* = 5–8 per group). Non-Tg males were set as 1 for relative mRNA units. Two-way ANOVA analysis with Tukey’s post hoc test: * = *p* < 0.05, ** = *p* < 0.01 versus same-sex non-Tg group.

**Figure 9 ijms-22-05365-f009:**
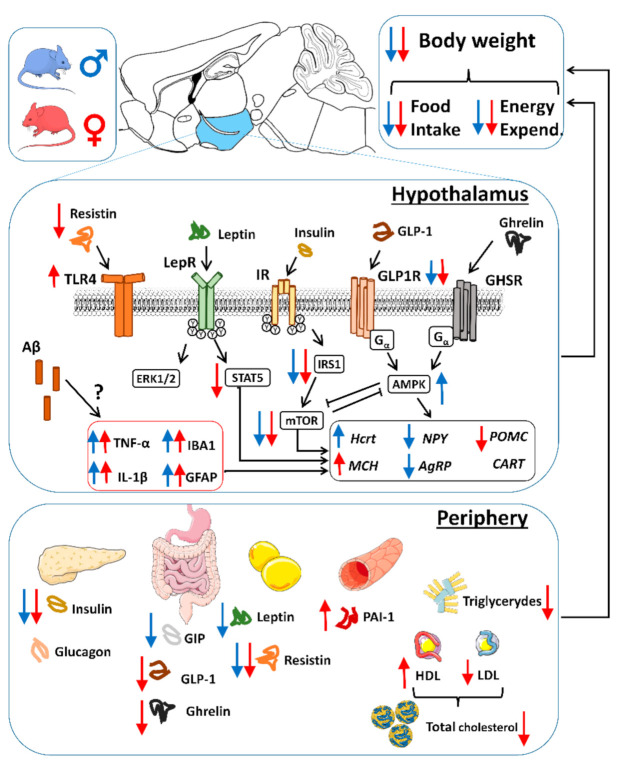
Schematic view of peripheral and hypothalamic alterations derived in negative energy balance and decreased body weight in 5xFAD mice. Increased or decreased observed levels are represented in blue for males and red for females. In the periphery, decreased insulin levels are associated with decreased GIP in males and GLP-1 in females, which are related to decreased body weight pattern. Low leptin levels were observed in males, whereas decreased ghrelin and resistin were more pronounced in females. Females also showed lower total triglyceride and total cholesterol levels, with increased HDL and decreased LDL, contributing to peripheral metabolic impairment. In the hypothalamus, insulin signaling was decreased in 5xFAD mice, accompanied by decreased leptin signaling and resistin hypothalamic levels in females. Despite absence of amyloid plaques, hypothalamic neuroinflammation was observed and contributed to hypothalamic dysfunction and lower body weight. Alterations in NPY/AgRP and orexin (Hcrt) in males seemed to contribute to decreased food intake, whereas females showed a tendency towards decreased overall neuropeptide expression but increased MCH levels. Hypothalamic and peripheral neuroendocrine dysfunction, which are sex-specific and aggravated in the full transgenic (increased Aβ burden) mice, are proposed as contributors to whole negative energy balance in 5RExFAD mice.

**Table 1 ijms-22-05365-t001:** Correlations between body weight and hypothalamic neuroinflammatory markers in 5xFAD mice at 6 months of age.

	Males			Females		
Body Weight Versus:	Non-Tg	Tg^5xFAD/-^	Tg^5xFAD/5xFAD^	Non-Tg	Tg^5xFAD/-^	Tg^5xFAD/5xFAD^
Il-1β (hyp)	ns ^1^	ns	ns	ns	R^2^ = 0.820 *p* = 0.034	R^2^ = 0.593 *p* = 0.025
TNF-α (hyp)	ns	ns	ns	R^2^ = 0.799 *p* = 0.016	R^2^ = 0.713 *p* = 0.071	R^2^ = 0.548 *p* = 0.035
GFAP (hyp)	ns	ns	ns	ns	ns	ns
IBA1 (hyp)	ns	ns	ns	R^2^ = 0.769 *p* = 0.021	R^2^ = 0.783 *p* = 0.046	R^2^ = 0.783 *p* = 0.019

^1^ ns, not significant.

**Table 2 ijms-22-05365-t002:** Correlations between body weight and hypothalamic and plasma hormones in 5xFAD mice at 6 months of age.

	Males			Females		
Body Weight Versus:	Non-Tg	Tg^5xFAD/-^	Tg^5xFAD/5xFAD^	Non-Tg	Tg^5xFAD/-^	Tg^5xFAD/5xFAD^
Resistin (hyp)	R^2^ = 0.584 *p* = 0.076	ns ^1^	ns	ns	R^2^ = 0.920 *p* = 0.009	R^2^ = 0.546 *p* = 0.036
Ghrelin (hyp)	ns	ns	ns	ns	ns	ns
Resistin (plasma)	R^2^ = 0.594 *p* = 0.072	ns	R^2^ = 0.704 *p* = 0.018	ns	R^2^ = 0.894 *p* = 0.001	R^2^ = 0.755 *p* = 0.011
Ghrelin (plasma)	ns	ns	ns	ns	ns	R^2^ = 0.580 *p* = 0.046
Insulin (plasma)	ns	R^2^ = 0.760 *p* = 0.023	R^2^ = 0.851 *p* = 0.003	ns	R^2^ = 0.708 *p* = 0.017	R^2^ = 0.578 *p* = 0.047
Leptin (plasma)	ns	R^2^ = 0.942 *p* =0.001	R^2^ = 0.612 *p* = 0.037	ns	ns	ns
GLP-1 (plasma)	ns	ns	ns	ns	ns	R^2^ = 0.622 *p* = 0.034
GIP (plasma)	ns	R^2^ = 0.813 *p* = 0.014	R^2^ = 0.644 *p* = 0.029	ns	ns	ns
PAI-1 (plasma)	ns	ns	ns	ns	R^2^ = 0.628 *p* = 0.033	R^2^ = 0.708 *p* = 0.017
Glucagon (plasma)	ns	ns	ns	ns	ns	ns

^1^ ns, not significant.

**Table 3 ijms-22-05365-t003:** Correlations between plasma insulin and insulin release-regulating hormones in 5xFAD mice at 6 months of age.

	Males			Females		
Insulin (Plasma) Versus:	Non-Tg	Tg^5xFAD/-^	Tg^5xFAD/5xFAD^	Non-Tg	Tg^5xFAD/-^	Tg^5xFAD/5xFAD^
GIP	ns ^1^	R^2^ = 0.712 *p* = 0.034	R^2^ = 0.572 *p* = 0.048	R^2^ = 0.659 *p* = 0.049	ns	ns
GLP-1	ns	ns	ns	ns	ns	R^2^ = 0.846 *p* = 0.003
Ghrelin	ns	ns	ns	ns	ns	ns
Glucagon	ns	ns	ns	R^2^ = 0.582 *p* = 0.077	ns	R^2^ = 0.495 *p* = 0.077

^1^ ns, not significant.

**Table 4 ijms-22-05365-t004:** Correlations between food intake-regulating hypothalamic neuropeptides and inflammatory markers IL-1β and TNF-α, as well as NPY in 5xFAD mice at 6 months of age.

	Males			Females		
	Non-Tg	Tg^5xFAD/-^	Tg^5xFAD/5xFAD^	Non-Tg	Tg^5xFAD/-^	Tg^5xFAD/5xFAD^
Il-1β versus						
NPY	ns ^1^	ns	ns	ns	ns	ns
AgRP	ns	ns	ns	ns	ns	ns
POMC	ns	ns	ns	ns	R^2^ = 0.211 *p* = 0.009	R^2^ = 0.546 *p* = 0.036
CART	ns	ns	R^2^ = 0.566 *p* = 0.051	ns	ns	ns
HCRT	ns	ns	ns	ns	R^2^ = 0.765 *p* = 0.052	ns
MCH	R^2^ = 0.868 *p* = 0.006	ns	ns	ns	ns	ns
Resistin (hyp)	ns	ns	ns	ns	R^2^ = 0.947 *p* = 0.005	R^2^ = 0.513 *p* = 0.045
Ghrelin (hyp)	ns	ns	ns	ns	ns	ns
**TNF-α versus**						
NPY	ns	ns	ns	R^2^ = 0.831 *p* = 0.011	ns	ns
AgRP	ns	ns	ns	ns	ns	ns
POMC	ns	ns	ns	ns	ns	ns
CART	ns	R^2^ = 0.829 *p* = 0.031	ns	ns	ns	ns
HCRT	ns	ns	ns	ns	ns	ns
MCH	ns	ns	ns	ns	ns	R^2^ = 0.436 *p* = 0.074
Resistin (hyp)	ns	R^2^ = 0.902 *p* = 0.013	R^2^ = 0.833 *p* = 0.004	ns	ns	R^2^ = 0.724 *p* = 0.007
Ghrelin (hyp)	ns	ns	ns	ns	ns	ns
**NPY versus**						
AgRP	ns	R^2^ = 0.945 *p* = 0.005	R^2^ = 0.896 *p* = 0.001	R^2^ = 0.707 *p* = 0.035	R^2^ = 0.952 *p* = 0.004	R^2^ = 0.985 *p* < 0.0001
POMC	ns	R^2^ = 0.707 *p* = 0.074	ns	ns	R^2^ = 0.797 *p* = 0.041	R^2^ = 0.640 *p* = 0.017
CART	ns	ns	ns	ns	R^2^ = 0.7987 *p* = 0.0409	ns
HCRT	ns	R^2^ = 0.780 *p* = 0.046	R^2^ = 0.692 *p* = 0.020	ns	ns	R^2^ = 0.511 *p* = 0.046
MCH	ns	R^2^ = 0.790 *p* = 0.043	R^2^ = 0.701 *p* = 0.018	ns	ns	R^2^ = 0.672 *p* = 0.012

^1^ ns, not significant.

**Table 5 ijms-22-05365-t005:** Correlations between cholesterol levels and triglyceride levels or glutamic oxaloacetic transaminase (GOT) in the plasma of 5xFAD mice at 6 months of age.

	Males			Females		
	Non-Tg	Tg^5xFAD/-^	Tg^5xFAD/5xFAD^	Non-Tg	Tg^5xFAD/-^	Tg^5xFAD/5xFAD^
Triglycerides versus						
Total cholesterol	ns ^1^	ns	ns	R^2^ = 0.912 *p* = 0.0002	R^2^ = 0.737 *p* = 0.013	R^2^ = 0.582 *p* = 0.010
HDL	ns	ns	ns	ns	R^2^ = 0.746 *p* = 0.012	R^2^ = 0.570 *p* = 0.011
LDL	ns	ns	ns	ns	ns	R^2^ = 0.761 *p* = 0.001
**GOT versus**						
Total cholesterol	ns	ns	ns	ns	ns	ns
HDL	ns	ns	ns	ns	ns	ns
LDL	ns	ns	ns	ns	ns	R^2^ = 0.426 *p* = 0.040

^1^ ns, not significant.

## Data Availability

The data that support the findings of this study are available on reasonable request from the corresponding author.

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
