# Peer review of "A Negative Energy Balance Is Associated with Metabolic Dysfunctions in the Hypothalamus of a Humanized Preclinical Model of Alzheimer’s Disease, the 5XFAD Mouse"

_ijms, 2021, doi:10.3390/ijms22105365_

Round 1
Reviewer 1 Report
This study demonstrated that the correlation between negative energy balance and metabolic dysfunction in the 5xFAD mouse.
- Authors must explain the acronyms the first time they appear in the text. (There are many overlaps.)
- In the figure 1A-F, authors need to show a quantitative graph for Aβ
- In the figure 5H, authors must change the representative picture.
Author Response
R1: This study demonstrated the correlation between negative energy balance and metabolic dysfunction in the 5xFAD mouse.
- Authors must explain the acronyms the first time they appear in the text. (There are many overlaps.)
Response: We checked and corrected acronyms appearing the first time in the text and eliminated overlaps.
- In the figure 1A-F, authors need to show a quantitative graph for Aβ.
Response: We added a quantitative graph for Aβ in hypothalamus and hippocampus (please, see new Figure 2G). Lines 187-189.
- In the figure 5H, authors must change the representative picture.
Response: We changed representative picture of blurred western blot bands (please, see Figure 5H).
Main changes are highlighted in yellow. English style has been checked.
Thank you.
Reviewer 2 Report
It is a very interesting manuscript where researchers study the role of the hypothalamus in a preclinical model of Alzheimer's disease. The manuscript and research is very well designed. The work is really, very complete.
Just comment on a few points:
Why haven't authors evaluated the role of GSK3beta instead of CDK5 on TAU? The insulin receptor regulates GSK3 beta.
Although they already mentioned it in the discussion, they should emphasize that although there are no plaques in the hypothalamus, it is the soluble amyloid that activates glial cells. This section is important and should be highlighted in the manuscript.
Please add these references in the manuscript:
The Involvement of Peripheral and Brain Insulin Resistance in Late Onset Alzheimer's Dementia.
Front Aging Neurosci. 2019 Sep 6;11:236. A metabolic perspective of late onset Alzheimer's disease. Pharmacol Res. 2019 Jul;145:104255.Author Response
R2: It is a very interesting manuscript where researchers study the role of the hypothalamus in a preclinical model of Alzheimer's disease. The manuscript and research is very well designed. The work is really, very complete.
Just comment on a few points:
- Why haven't authors evaluated the role of GSK3beta instead of CDK5 on TAU? The insulin receptor regulates GSK3 beta.
Response: We agree with the reviewer comment. Because insulin signaling in the hypothalamus was decreased, we also evaluated the role of GSK3-β and observed low hypothalamic GSK3-β activity in 5xFAD mice (please, see the new Figure S4: the enhanced ratio of inhibitory serine phosphorylation/activatoy tyrosine phosphorilation). Although it is true that insulin targets, such as mTOR and GSK3-β, have been previously related to Tau hyperphosphorylation, other kinases, such as CDK5, have been demonstrated to contribute to Tau hyperphosphorylation. Our results suggest that the observed increased Tau phosphorylation in the hypothalamus of 5xFAD mice seems to be related to CDK5 and neuroinflammation and not to GSK3-β overactivation. Lines 827-831.
- Although they already mentioned it in the discussion, they should emphasize that although there are no plaques in the hypothalamus, it is the soluble amyloid that activates glial cells. This section is important and should be highlighted in the manuscript.
Response: We thank the reviewer for the suggestion. There has been extensive research suggesting that soluble Aβ oligomers, which we cannot be observed by immunohistochemistry, can lead to activation of microglia, brain insulin resistance and Tau hyperphosphorylation. As we now depicted in the manuscript, this should be highlighted as the possible cause for neuroinflammatory status observed in 5xFAD mice in the apparent absence of Aβ deposition in the hypothalamus. Further analysis using methodologies oriented to measure those oligomers will clarify this fact. In any case, since glial cells are main regulators of Aβ levels in the brain and mediate Aβ clearance in the brain, the presence of activated microglia might indicate a neurofinflammatory response triggered by soluble Aβ that precludes it deposition. Lines 662-668.
- Please add these references in the manuscript: The Involvement of Peripheral and Brain InsulinResistance in Late Onset Alzheimer's Dementia. Folch J, Olloquequi J, Ettcheto M, Busquets O, Sánchez-López E, Cano A, Espinosa-Jiménez T, García ML, Beas-Zarate C, Casadesús G, Bulló M, Auladell C, Camins A.Front Aging Neurosci. 2019 Sep 6;11:236. A metabolic perspective of late onset Alzheimer'sdisease. Ettcheto M, Cano A, Busquets O, Manzine PR, Sánchez-López E, Castro-Torres RD, Beas-Zarate C, Verdaguer E, García ML, Olloquequi J, Auladell C, Folch J, Camins A. Pharmacol Res. 2019 Jul;145:104255.
Response: We included the references (refs. 42 and 62) and thank the reviewer for the suggestions.
Thank you